# Molecular Chaperones in Cancer Stem Cells: Determinants of Stemness and Potential Targets for Antitumor Therapy

**DOI:** 10.3390/cells9040892

**Published:** 2020-04-06

**Authors:** Alexander Kabakov, Anna Yakimova, Olga Matchuk

**Affiliations:** Department of Radiation Biochemistry, A. Tsyb Medical Radiological Research Center—Branch of the National Medical Research Radiological Center of the Ministry of Health of the Russian Federation, Koroleva 4, Obninsk 249036, Russia; anna.prosovskaya@gmail.com (A.Y.); matchyk@mail.ru (O.M.)

**Keywords:** stem cell phenotype, epithelial-to-mesenchymal transition (EMT), heat shock protein (HSP), glucose-regulated protein (GRP), heat shock factor 1 (HSF1), tumor necrosis factor receptor-associated protein 1 (TRAP1), immunophilin, protein disulfide isomerase

## Abstract

Cancer stem cells (CSCs) are a great challenge in the fight against cancer because these self-renewing tumorigenic cell fractions are thought to be responsible for metastasis dissemination and cases of tumor recurrence. In comparison with non-stem cancer cells, CSCs are known to be more resistant to chemotherapy, radiotherapy, and immunotherapy. Elucidation of mechanisms and factors that promote the emergence and existence of CSCs and their high resistance to cytotoxic treatments would help to develop effective CSC-targeting therapeutics. The present review is dedicated to the implication of molecular chaperones (protein regulators of polypeptide chain folding) in both the formation/maintenance of the CSC phenotype and cytoprotective machinery allowing CSCs to survive after drug or radiation exposure and evade immune attack. The major cellular chaperones, namely heat shock proteins (HSP90, HSP70, HSP40, HSP27), glucose-regulated proteins (GRP94, GRP78, GRP75), tumor necrosis factor receptor-associated protein 1 (TRAP1), peptidyl-prolyl isomerases, protein disulfide isomerases, calreticulin, and also a transcription heat shock factor 1 (HSF1) initiating HSP gene expression are here considered as determinants of the cancer cell stemness and potential targets for a therapeutic attack on CSCs. Various approaches and agents are discussed that may be used for inhibiting the chaperone-dependent development/manifestations of cancer cell stemness.

## 1. Overviewing Introduction: Cancer-Associated Stemness, Its Molecular Basis, and a Role in the Pathogenesis

The terms “cancer stem cells” (CSCs) and “stemness” as well as the respective concept emphasizing a role of CSCs in malignant growth appeared in oncology decades ago [1,2]. At present, it is quite reliably established that CSCs or CSC-like cells are involved in the development of most human malignancies and responsible for such key events in the cancer pathogenesis as invasive tumor progression, tumor recurrence, and the spread of metastases. Being more chemoresistant and more radioresistant than non-stem cancer cells, CSCs may survive after chemo- and radiotherapy and thus be a cause of post-treatment cancer relapses [1,2,3,4,5]. There is a set of specific features that characterize a minor (but extremely dangerous) fraction of cancer cells as CSCs and, of course, a set of molecular determinants defining such features is known. In the next three subsections, the most characteristic properties of CSCs and the main ways of forming the CSC phenotype are considered in the context of cancer pathogenesis.

### 1.1. Tumorigenicity of CSCs and Maintaining the Cancer Stemness

In several points, CSC phenotypes are similar to those of normal stem cells (SCs); the latter play a key role in organ development during embryogenesis and also ensure tissue homeostasis or tissue regeneration in adult organisms [6]. The most characteristic properties of both SCs and CSCs are their capacity for self-renewal via cell division (symmetric or asymmetric) and their huge proliferative potential coupled to resistance to replicative senescence. Moreover, the same signaling pathways, namely Wingless-related integration site (Wnt)/β-catenin, Hedgehog, and Notch, act in both normal SCs and CSCs [1,2,3,4,5]. The transcription factors NANOG, octamer-binding transcription factor 4 (OCT4) and sex determining region Y-box 2 (SOX2) [1,2,3,4,5], Cripto-1, a member of the epidermal growth factor (EGF)–cripto-FRL1-cryptic (CFC) family and a co-receptor for transforming growth factor-β (TGF-β) ligands [7], and an epigenetic regulator, such as PR domain-containing protein M14 (PRDM14) [8], are known to maintain the stemness in both embryonic SCs and CSCs. Nevertheless, tumor-initiating CSCs differ from non-cancerous SCs [1,2,3,9]. 

In vivo, the appearance of CSCs is supposed to be a result of oncogenic mutations in the genome of normal SCs or progenitors [2,5], and therefore such mutated cells have some genetic and epigenetic changes that fatally dysregulate cellular signal transduction networks and gene expression, thereby making CSCs what they are. It is well known that CSCs have a number of phenotypic markers, including specific cell surface antigens and upregulated expression/activities of certain enzymes and membrane transporters which enable researchers to isolate the CSC fractions from heterogenic cell populations [1,2,3,4,10]. However, the most critical feature of CSCs is their tumorigenicity, i.e., an ability to initiate malignant tumors in vivo, and this feature is defined by a set of genetic and epigenetic factors reprogramming cell division regulation, cellular responses and signaling pathways, gene transcription, and metabolism toward the tumorigenic stem cell phenotype that ensures growth of each malignancy (for a review, see [1,2,3,4,5]). 

The intrinsic capacity of CSCs to their endless self-renewal via mitotic division is, undoubtedly, one of the factors contributing to the tumorigenicity. A predominance of the symmetric cell division (generating two stem cell daughters) over the asymmetric division (generating one stem cell daughter and one non-stem, usually more differentiated cell) is thought to be responsible for the increase in CSC pools and one of the determinatives of malignant growth. It is generally accepted that the transcription factors NANOG, OCT4, and SOX2 (as well as their stabilizers and activators) contribute to the self-renewal of CSCs [1,2,3,4,5]. Besides, the CSC capacity for endless renewal is causally linked to the high activity of telomerase inside them, an enzyme that joins terminal nucleotide sequences to the 3’ ends of telomeres; this activity saves proliferating CSCs from the critical shortening of chromosomal telomeres that otherwise leads to replicative senescence or cell death [3,11]. Autophagy appears to be an additional mechanism that allows CSCs to renew themselves by intracellularly digesting aberrant macromolecules and aged organelles, and as a result to evade apoptosis and senescence [12]. Apparently, the self-renewal of CSC populations plays an extremely important role in cancer pathogenesis because it is what ensures, first, unlimited expansive growth of malignancies in vivo and, second, constant reproduction of the pool of tumorigenic malignant cells with high adaptive potential and enhanced resistance to therapeutics that may lead to post-treatment tumor recurrence and poor patient outcomes.

Certain signaling networks and epigenetic factors, being deregulated in CSCs, contribute to the self-renewal of CSCs and maintenance of their tumorigenic stem phenotype. So, it was revealed that deregulated Notch signaling and abnormally activated Hedgehog signaling promote self-renewal of CSCs in malignancies with different origins and localization by ruling the expression of known modulators and markers of the stemness, such as Slug, Twist, SOX2, B lymphoma Mo-MLV insertion region 1 homolog (BMI1), and OCT4 [1,2,3,4,5,9]. In its turn, the Wnt/β-catenin signaling pathway, hyperactivated in CSCs, was shown to upregulate the expression of CD44, CD133 or Prominin 1 (PROM1), leucine-rich repeat-containing G-protein coupled receptor 5 (LGR5), aldehyde dehydrogenase (ALDH), and the membrane transporters ABCB4 and ABCG2 (all are phenotypic markers of cancer cell stemness) (Figure 1; [1,2,3,4,5,9]). Moreover, Wnt/β-catenin signaling is involved in the so-called epithelial-to-mesenchymal transition (EMT) and in cross-talk with other signaling pathways, such as Notch, Hedgehog and TGF-β/bone morphogenic protein (BMP) pathways; all this contributes to the CSC-associated tumorigenicity and occurrence of distant metastases [1,2,3,4,5]. 

### 1.2. EMT and Other Factors Affecting the CSC Phenotype

Most human malignancies are carcinomas (i.e., derive from the epithelium), and EMT results in a transformation of non-stem carcinoma cells into fibroblast-like CSCs expressing N-cadherin instead of E-cadherin and vimentin instead of cytokeratin; this switching is regulated by Slug, Snail, Twist, zinc finger E-box-binding homeobox 1 and 2 (ZEB1/2) [1,2,3,4,5], the oncogene multiple copies in T-cell malignancy 1 (MCT-1/MCTS1) product [13], and a zinc finger protein PRDM14 [8,14]. The EMT-associated phenotypic modulation alters gene profile expression, cell metabolism, cell polarity, and cell–cell and cell–matrix interactions, thus promoting the enhanced motility/migration of CSCs and formation by them of multicell spheroids, perivascular niches, and distant metastases. Having undergone EMT, fibroblast-like CSCs produce greater amounts of angiogenic factors (vascular endothelial growth factor or VEGF and others) and this leads to stronger stimulation of cancer-related angiogenesis and, consequently, accelerated tumor vascularization and tumor expansion with forming metastases. It is generally accepted that EMT and EMT-generated CSCs play a pivotal role in the dissemination of metastases. Besides, those EMT-generated highly mobile CSCs form the invasive front of growing solid tumors [2,3,5]. Integrin β1 (one of the biomarkers of cancer stemness) expression [15] and activation of extracellular proteinases [16] facilitate the migration of CSCs across the extracellular matrix and their invasion into surrounding normal tissues and nearby organs as well as their penetration into blood and lymphatic vessels to spread metastases. All these phenotypic alterations are certainly associated with the increase in tumorigenic potential. It should be noted that part of the CSCs generated by EMT may enter a state of quiescence and then sojourn in it without mitotic division and without apoptosis or senescence. Such quiescent apoptosis-resistant CSCs are less sensitive to cytotoxic treatments than actively proliferating non-stem cancer cells; this is one of the reasons why CSCs can survive better after chemotherapy and radiotherapy [2,3,4,5]. 

In malignancies of non-epithelial or mesenchymal origin, EMT-resembling changes in tumorous cell phenotypes also occur, which transform part of the differentiated non-stem cancer cells into dedifferentiated CSC-like cells with a higher capacity for migration, invasion, and metastasis formation [2,3,4,5,17]. As well as the symmetric CSC division, both EMT and similar (dedifferentiating) phenotypic transitions in tumor cell populations are a way of increasing the pool of CSCs that are highly resistant to therapeutics and yield invasive tumor growth and development of metastases; therefore, these mechanisms are extremely important for cancer pathogenesis (Figure 1). The reverse modulation of cancer cell phenotypes, namely mesenchymal-to-epithelial transition (MET), can occur afterward within distant metastases under the action of differentiating factors, which retransform fibroblast-like CSCs into typical carcinoma cells with tightly apposed cell–cell contacts and higher proliferative activity. Such alternating EMT and MET promotes multiple and accelerated growth of solid carcinomas in various organs, which aggravates the pathogenesis [1,18]. An additional way of accumulating non-stem cancer cells due to CSCs is the asymmetric division of the latter. The reversible transitions between non-stem cancer cells and CSCs are schematically shown in Figure 1, with indication of some extracellular and intracellular factors setting such a phenotypic plasticity.

In vivo, the induction of EMT and similar phenotypic transitions increasing CSC amounts occur upon the influence of various components of the tumor microenvironment that form the so-called CSC niche [2,3,4,5]. In aggressively growing carcinomas, EMT can be triggered by a set of microenvironment-derived factors, such as hormones, growth factors, cytokines, and other bioactive molecules, including platelet-derived growth factor (PDGF), TGF-β, tumor necrosis factor-α (TNF-α), fibroblast growth factor (FGF), interleukins 6 and 8 (IL6, IL8), C-X-C motif chemokine ligands 5, 7 and 12 (CXCL5, CXCL7, CXCL12), Tenascin C, hyaluronic acid, nitric oxide, and others (see Figure 1). These CSC niche-associated biomolecules come in from nearby lymphatic and blood vessels, adjacent zones of inflammation and edema, noncancerous stromal cells (cancer-associated fibroblasts or CAFs, mesenchymal stem cells, macrophages, vascular cells), and the extracellular matrix. It is important to note that CSCs secrete cytokines and growth factors, which, in turn, manipulate surrounding normal cells to recruit them for the CSC niche’s formation/maintenance and tumor progression (reviewed in [2,3,4,5]). Notably, some of the EMT- and cancer stemness-promoting biomolecules can be “packed” into special vesicles (exosomes), which are secreted and entrapped by different cells within the CSC niche. Those exosomes can contain TGF-β2, TNF1α, IL6, Akt (protein kinase B), β-catenin, matrix metalloproteinases (MMPs), microRNAs, and other EMT-inducing factors that influence tumor cell phenotypes and upregulate stemness-associated properties, such as motility, extracellular proteinase activity, invasiveness, and spheroid formation [19]. Consequently, such exosomes are thought to be important inducers of EMT and the CSC-like phenotype. 

Additional EMT-inducing stimuli are hypoxia, nutrient deprivation, and acidosis (low pH) in the microenvironment of some cancer cells locally undergoing such stressful conditions in tumor regions with inadequate vascularization and failed blood circulation, which are typical for aggressive solid malignancies (for further details, see [1,2,3,4,5,17]). As compared with exosomes secreted under normoxic conditions, hypoxia-stressed tumor cells secrete exosomes that exhibit higher MMP activity, are more loaded with EMT-triggering molecules, and, therefore, are more conducive to cancer stemness development [20]. All those external inducers of EMT activate intracellular Hedgehog, Wnt, and Notch signaling pathways along with Cripto-1-mediated signaling, and also transcription-regulating factors HIF1α, HIF2α, signal transducer and activator of transcription 3 (STAT3), NANOG, OCT4, heat shock factor 1 (HSF1), and nuclear factor κB (NFκB) that contribute to cancer cell stemness development and the formation of the CSC niche [1,2,3,4,5,7,8,17]. MicroRNAs, being epigenetics factors of cellular regulation, can also modulate the processes of EMT and regulate the formation of CSC-like phenotypes (reviewed in [3]). The MCT-1/miR-34a/IL-6/IL-6R signaling axis was revealed to promote EMT and cancer stemness in triple-negative breast cancer [13]. 

Importantly, both drug treatment and radiation exposure and immunotherapy may augment the pools of CSC-like cells resistant to therapeutics. It was found that γ-irradiation at doses 2–10 Gy increased the proportion of CSC-like radioresistant SP cells in murine B16 melanoma [21]. Radiation-induced EMT was described for breast cancer [22], esophageal squamous carcinoma [23], and colorectal carcinoma [24]. Additionally, the antitumor drug gemcitabine was demonstrated to induce urokinase-type plasminogen activator (uPA)-mediated invasion, spheroid formation, and resistance to apoptosis in CSC-like cells from pancreatic carcinoma [25]. If such mechanisms are realized in target tumors, it may lead to a failure of radiotherapy and chemotherapy. Notably, immunotherapy based on the action of tumor-reactive cytotoxic T-lymphocytes (CTLs) may stimulate CTL-mediated selection/evolution of cancer cell phenotypes toward CSC-like phenotypes, forming immune-refractory tumors [26,27].

### 1.3. Resistance of CSCs to Therapeutics, Immune Attack, and Stressful Conditions

CSCs, arising as a result of EMT or similar phenotypic modulations, acquire a number of stemness-associated properties that allow CSCs to withstand various cytotoxic exposures and evade elimination through immunogenic cell death. One of the characteristic signs of CSCs is their relatively high resistance to chemotherapeutic drugs, which is partly explained by the enhanced expression/activity of the membrane transporters of the ATP-binding cassette (ABC) family [2,3,4,5]. Those ABC transporters are transmembrane protein complexes that hydrolyze ATP and pump cell-entering xenobiotics out of the cell; such transporters play a key role in the phenomenon of multidrug resistance, which is a serious challenge for anticancer chemotherapy [28]. As compared with non-stem cancer cells, CSCs exhibit much more active work of the reserpine- or verapamil-sensitive ABCB1 and ABCG2 transporters, which enables the use of flow cytometry for identification and isolation of the CSC fractions as cells more effectively excluding Hoechst 33342 dye or Rhodamine 123 dye (the so-called “side population” of less stained cells or SP cells) [1,3,10]. Overexpression of the ABCB1 transporter (often referred to as the MDR1 gene product: P-glycoprotein or gp170) in CSCs is thought to be due to hyperactivated Hedgehog signaling [3,10]. Besides, this appears to be connected to the activation and action of a heat shock transcription factor 1 (HSF1) because, like promoter regions of all heat shock-genes, the MDR1 gene promoter region contains a heat shock element (HSE)—the specific nucleotide sequence that is recognized by the activated HSF1 to initiate a respective gene transcription [29] (see Section 2 for further details). The EMT-associated Wnt/β-catenin signaling pathway ensures the overexpression of ABCG2 transporters, which save CSCs from cytotoxic anticancer drugs, such as doxorubicin, imatinib, and others [3,30]; so, the ABC transporters overexpressed in CSCs are important molecular determinants of the stemness and one of the causes of the chemoresistance intrinsic to CSCs.

Besides their well-known chemoresistance, CSCs possess enhanced resistance to the cytotoxic action of ionizing radiation, and this is an additional challenge for cancer treatment because CSCs may remain viable after radiotherapy, resulting in malignant growth recurrence. One of the factors promoting the high radioresistance of CSCs is their lower proliferative activity as compared with non-stem cancer cells; being a product of EMT, a part of the CSC pool may transiently be in a state of quiescence (see above) that makes them more radioresistant. The high telomerase activity in CSCs [3,11] may help them to escape from post-radiation replicative senescence. Moreover, CSCs have improved DNA damage response mechanisms that include the expression of Snail, excision repair cross-complementation group 1 (ERCC1) and nibrin or NBS1 – proteins assisting DNA repair in CSCs after ionizing radiation or genotoxic drug exposures (see [3] for a review). In addition to the highly active telomerase and better activated DNA repair systems, dysfunction of the apoptosis-regulating tumor suppressor p53 [3] and the enhanced expression of antiapoptotic proteins survivin [31] and B-cell lymphoma 2 (Bcl-2) [32] can also contribute to the increased radioresistance of CSCs; analogous cytoprotective mechanisms can save CSCs from some DNA-damaging drugs like cisplatin, bleomycin, etoposide, and others.

A failing immune attack against neoplastic cells is one of the essential conditions for oncogenesis. In this respect, tumorigenic CSCs are a very poor target for the host immunity, which appears to be due to specific regulation of their stem phenotype [5,9]. The downregulated expression of major histocompatibility complex (MHC) class I molecule, low molecular weight protein (LMP), transporter associated with antigen processing (TAP), and Toll-like receptor 4 (TLR4) was revealed in CSCs, which preserves them from recognition by the innate immune system and T cell attack (reviewed in [5,9]). Additionally, CSCs are able to affect the tumor microenvironment so that the immune response against the tumor is impaired. In particular, CSCs produce immunosuppressive factors, such as IL4 and TGF-β; the latter causes arrest of the proliferation of active T cells, inactivation of natural killer (NK) cells, and cancer-promoting M2 macrophage polarization [5]. IL6, one of the mediators of the immune/inflammation response and inducers of EMT, can increase cancer-associated stemness and MMP activity and M2 macrophage polarization via the MCT-1/miR-34a/IL-6/IL-6R signaling pathway [13]. The extracellular (secreted) proteases, such as MMP2, MMP9, and uPA, may split some immune system-recognizable proteins at the CSC surface, thus helping CSCs to hide from antitumor immunity. CSCs have specific mechanisms to recruit and interact with M2 macrophages [5,13]. SOX2 and DKK1 expressed in CSC-like cells allow them not to be eliminated via NK cell-mediated clearance [33]. Importantly, CTL-tumor interactions (including those induced by immunotherapy) may trigger the immune selection of malignant cells along with evolution of their phenotypes toward CSC-like phenotypes resistant to CTL-mediated killing [26]. Such phenotypic evolution was shown to be driven by the NANOG/TCL1A/Akt pathway [26,27]; a role of the chaperone HSP90A in this CSC-generating mechanism is considered in the Section 3.1.1. All this testifies the capacity of CSCs for (i) evasion of immune attacks and immunogenic cell death, (ii) multitarget suppression of the antitumor immune response and (iii) remodeling the host immune system in favor of accelerated growth and metastatic spread of a malignancy.

Among the distinct features of CSCs there is their elevated resistance to oxidative stress, hypoxia and nutrient deficiency. The generation and levels of reactive oxygen species (ROS) are known to be relatively low in CSCs. The hypoxia-activated HIF1α signaling pathway in CSCs ensures the decrease in their endogenous production of ROS [3,5]. An additional cause of the low ROS level in CSCs and their improved protection against free radicals or (pro)oxidants may be the enhanced expression of certain enzymes such as catalase, superoxide dismutase 2 (SOD2), peroxiredoxin 3 and aldehyde dehydrogenase (ALDH) (see [3] for a review). ALDH is a generally accepted biomarker of CSCs that can be used for typing and isolation of the CSC fractions. The high levels of ALDH expression/activity were found in CSCs of different origin and this also contributes to the increased radio- and chemoresistance of CSCs [3,10]. The fact that CSCs can arise and reside in the hypoxic tumor regions with poor vascularization and inadequate blood supply suggests a capacity of these cells to effectively adapt to energetically unfavorable conditions. Indeed, in the context of oxygen deprivation and restricted nutrient availability, CSCs rearrange their energy metabolism toward the domination of glycolysis (the so-called Warburg effect) with more active uptake and consummation of glucose to compete with other cells within the hypoxic niche. CSCs may have the high-affinity glucose transporter type 3 (GLUT3) expression that gives them advantages in the hypoxic niche-associated conditions of starvation and glucose deficiency [3,34]. Notably, AMP-activated kinase (AMPK) is a cellular energy sensor which modulates metabolism in cells lacking nutrients [35]. There are reports indicating an involvement of AMPK in mechanisms adapting CSCs to hypoxia and nutrient limitation [36,37]. The other energy metabolism-related enzyme, hexokinase 2 (HK2), functions in CSCs as an activator of glycolysis and anti-apoptotic factor conferring chemoresistance [38]. Other (HSF1- and chaperone-related) adaptive responses of CSCs to hypoxia, starvation and hypoglycemic stress are analyzed in the Section 3 and Section 4

All the listed peculiarities of CSCs characterize these cells as key drivers of tumorigenesis and an extremely serious challenge for cancer treatment. It is CSCs that are responsible for emergence of immune-refractory tumors, unlimited tumor expansion, distant metastasis formation, and tumor recurrence after therapy; obviously, there is a great need for the creation of therapeutic modalities to oppose CSCs. The present review reveals a role of molecular chaperones (protein regulators of polypeptide chain folding) in formation of the CSC phenotype and CSC niche, and also in mechanisms of the CSC resistance to chemotherapy and radiotherapy. Various approaches to targeting of the activity/expression of chaperones in CSCs are here overviewed as potential ways for overcoming the cancer stemness-associated problems in the treatment of malignancies. Numerous publications on the relevant topic hopefully indicate that suitable chaperone-targeting agents may, if not kill CSCs, either prevent phenotypic modulations/evolution leading to their accumulation or make existing CSCs to lose their special properties and turn into non-stem cancer cells which are more vulnerable to antitumor therapeutics and immune attack (such opportunities are denoted in Figure 1).

## 2. Molecular Chaperones: Localization, Activities, and Implication in Cellular Stress Responses

The term “molecular chaperones” designates a certain class of cellular proteins that regulate the folding of polypeptide chains and conformational transitions in molecules of various client proteins. Such an activity allows ubiquitous chaperones to control biosynthesis, maturation, and degradation of proteins, protein transport across membranes, assembly of protein oligomers, conformation-dependent functions of different enzymes, receptors, transcriptional factors, etc and hence be important modulators of virtually all vital processes in the cell [39]. 

Many molecular chaperones are the so-called ‘heat shock proteins’ (HSPs) or their analogues. According to the kDa values of their molecular mass and structural homology, the major chaperones can be divided into six subfamilies: HSP110 (HSPH), HSP90 (HSPC), HSP70 (HSPA), HSP60/HSP10 (HSPD/E), HSP40 (DNAJ), and the “small” (20–27 kDa) HSPs (HSPB) [39,40,41]. The cytosol of mammalian cells contains rather large amounts of HSP90, HSP70, HSP40, and HSP27. A 75 kDa chaperone, tumor necrosis factor receptor-associated protein 1 (TRAP1) or HSPC5 [41], is a member of the HSP90 subfamily and is localized in mitochondria [42]. HSP60 belongs to the class of ‘chaperonins’ and is mainly present in mitochondria, where it works together with HSP10 [39,40].

The other family of molecular chaperones unites ‘glucose-regulated proteins’ (GRPs): GRP75 (mortalin, an analogue of cytosolic HSP70), which is localized to mitochondria; and GRP94, GRP78, (BiP) and GRP170 (three analogues of cytosolic HSP90, HSP70, and HSP110, respectively), which are preferentially localized to the endoplasmic reticulum (ER), although some part of GRP78 is present in the cytosol, nucleus, mitochondria, on the cell surface, and secreted [43]. 

Calreticulin [44,45] and calnexin [46] are Ca^2+^-binding chaperones preferentially localized to the ER and sometimes on the cell surface.

Protein disulfide isomerases are chaperones belonging to the thioredoxin superfamily; these enzymes act as thiol-disulfide oxidoreductases and disulfide isomerases by forming or breaking intramolecular S-S bonds between cysteine residues in proteins, which affects the folding and stability of protein globules [47]. The major pool of protein disulfide isomerases is in the ER, but these chaperones can also reside in the cytosol, nucleus, mitochondria, and on the cell surface or be secreted [47].

Immunophilins are ubiquitous peptidyl-prolyl isomerases and make up a separate group of chaperones regulating the bends of polypeptide chains at sites of the L-proline location [48]. Immunophilins can act in cooperation with the HSP70/HSP90 chaperone machine (see Figure 2 and [49,50]).

HSP110, HSP90, TRAP1, HSP70, HSP60, GRP170, GRP94, GRP78, and GRP75 contain ATP-binding domains and exhibit ATP-ase activity that is critically important for performing the chaperone function [39,40]. HSP40, HSP27, HSP10, immunophilins, protein disulfide isomerases, calreticulin, and calnexin do not possess the ATP-ase activity. 

In the cytosol of the unstressed cell, the constitutively expressed HSP70 and HSP90 are involved in the maturation and stabilization of protein molecules, proteolysis, and intracellular signal transduction [39,40]. Under normal conditions, the main substrates of those HSPs are incomplete polypeptide chains and various premature client proteins, whose maturation or activity regulation needs such chaperoning. The HSP70/HSP90 chaperone machine consumes ATP and recruits a set of (co-)chaperones and co-factors, such as HSP40, Hip, Hop, p23, Cdc37, immunophilins, and CHIP, which regulate the ATP-ase activities and ATP/ADP exchange in ATP-binding domains of HSP70 and HSP90 or assist to fit client protein molecules to the substrate-binding domains of HSP70 and HSP90 and then catalyze the proper folding (structural maturation) of bound client proteins or otherwise lead them to ubiquitination and degradation (Figure 2). Importantly, inhibition of the ATP-ase activity of intracellular HSP90 (e.g., by cell-permeable inhibitors) may lead to CHIP-mediated ubiquitination and proteasomal degradation of bound client proteins instead of their maturation and stabilization (the upper path in Figure 2); such an approach is considered as a promising strategy for repressing tumors and sensitizing them to therapeutics [51]. Cytosolic HSP110 is an HSP70-related chaperone that assists HSP70 in the primary interactions with protein substrates [39]. The small chaperone HSP27 is involved in intracellular signal transduction and maintenance of cellular redox homeostasis [52]. Chaperones GRP170, GRP94, GRP78, GRP75, calreticulin, and calnexin rule the maturation, processing, and transport of proteins within the ER, while HSP60/HSP10 and TRAP1 (HSP75) perform similar functions within mitochondria. Some molecules of GRP170, GRP94, GRP78, protein disulfide isomerases, and calreticulin are exposed on the cell surface or exported into the extracellular space; those cell surface-localized or secreted chaperones can promote cell surface-exposition or secretion of certain (glyco)proteins and peptides [43,44,45,46,47,48].

In the case of proteotoxic stress (heat shock, low pH, hypoxia, energy deprivation, reactive oxygen species (ROS) burst, heavy metal ions, others), many cellular proteins undergo unfolding or misfolding and then aggregate with each other. When the cytoplasm is overloaded by stress-damaged protein molecules, the cytosolic chaperones HSP90, HSP70, and HSP27 can attenuate protein aggregation by binding to unfolded and misfolded proteins to maintain them in a folding-competent state or, alternatively, ensure their degradation. Then, the ATP-dependent chaperone machinery helps the stressed cell to get rid of all damaged proteins and protein aggregates by catalyzing the refolding or degradation of denatured protein molecules (see Figure 2 and [39,40,50]). It is known that HSPs participate in the regulation of proteolysis and autophagy, which may favor the post-stress cell’s recovery [53]. Besides their ability to attenuate stress-associated proteotoxicity, both HSP90 and HSP70 and HSP27 are implicated in cytoprotection by blocking apoptotic pathways in stressed cells [54].

In mammalian cells, HSP110, HSP90, HSP70, HSP60, HSP40, and HSP27 are expressed constitutively, but their inducible forms are products of the transcriptional ‘heat stress response’, which is triggered by a heat shock transcription factor 1 (HSF1) whose activation results from the denaturation of intracellular proteins after any proteotoxic exposure (see Figure 3 and [55]). In vivo, the proteotoxic effects and HSF1 activation may take place upon such pathophysiological states as hypoxia (ischemia), acidosis, inflammation, edema, and others. It is generally accepted that the stress-induced accumulation of inducible HSPs contributes to cell survival/recovery and confers transient (excess HSP-based) tolerance to the next stresses of different types. This HSF1-dependent induction of HSPs is one of the major stress-responsive cytoprotective mechanisms that defines the capacity of HSP-enriched mammalian cells to quickly adapt to various stresses or pathophysiological states [55].

Unfortunately, all the cytoprotective activities of HSF1 and HSPs are manifested toward cancer cells as well. The upregulated expression of inducible HSP90, HSP70, or HSP27 is found in many human malignancies and is often associated with poor tumor response to therapy and poor outcome for patients [40]. Some part of HSP90 and HSP70 can be exposed on the surface of cancer cells or secreted out; these cancer cell surface-associated HSPs and extracellular HSPs also play a role in the cancer pathogenesis [56]. It was established that increased HSP levels contribute to malignant growth and may preserve drug- or radiation-treated cancer cells from apoptosis, necrotic death, autophagic death, senescence, and mitotic catastrophe (reviewed in [40,57]). Consequently, HSPs and HSF1 are regarded as attractive molecular targets for antitumor therapeutics [57,58]; new combinatorial strategies on the development of highly selective and low-toxicity inhibitors of HSPs are now used on the basis of computer-aided drug design [59].

The so-called “ER stress” provoked by hypoglycemia, hypoxia, acidosis, Ca^2+^ imbalance, inhibition of N-glycosylation, or other stressing exposure leads to the failure of proper protein folding within the ER compartment, thereby triggering the “unfolded protein response” (UPR) and GRP induction (Figure 4 and [43,46]). The newly expressed (excess) GRPs may help the stressed cell to carry out ‘protein quality control’ by catalyzing the renaturation or degradation of stress-damaged protein molecules to prevent protein aggregation in the ER and mitochondria. Besides, GRPs are intracellular regulators of signaling pathways, Ca^2+^ homeostasis, apoptosis and autophagy, protein transport and secretion, immune and inflammation responses, etc [43,46]. Similar to what tumorous HSPs do, GRPs in cancer cells promote both malignant growth and tumor resistance to therapeutics [43]. Therefore, GRP-targeting agents are thought to be potential tools for repressing human malignancies or sensitizing them to chemotherapy and radiotherapy (see Section 4).

Protein disulfide isomerases are obligatory players in the UPR and cell recovery after ER stress [46,47]. As the redox-sensitive intramolecular disulfide bonds are extremely important for protein folding and stable configuration design of protein globules, protein disulfide isomerases perform the function of redox-dependent chaperones. Protein disulfide isomerases, together with other chaperones and co-chaperones, can participate in the refolding of stress-damaged protein molecules or promote the degradation of misfolded proteins [46,47].

Calreticulin is also an ER-associated chaperone: This Ca^2+^-binding lectin-like protein is involved in the quality control process during the synthesis, maturation, and transport of many secretory and membrane proteins, including integrins, surface receptors, glucose transporters, and others. In addition, calreticulin regulates the Ca^2+^ homeostasis, mRNA stability, cell adhesion/migration, phagocytosis, etc. [44,45,46]. Calreticulin-encoding gene (CALR) expression can be activated by a set of transcription factors in response to ER stress or calcium depletion [44,45].

## 3. Contribution of HSPs, HSF1, and Immunophilins to the Regulation of the CSC Phenotype, and Approaches to Overcoming Chaperone- or HSF1-Conferred Cancer Stemness

Taking into account that molecular chaperones are of extreme importance for cell vitality and cell differentiation [39], it would be logical to expect that they are actively implicated in the regulation of the stem phenotype in both normal SCs and CSCs. There are summarized data indicating that HSPs are indeed necessary for the self-renewal and survival of normal SCs [60]. Lettini et al. [61] described the contribution of several HSPs to CSC maintenance. In the present subsection, the same problem, namely the role of major HSPs in forming and maintaining the CSC phenotype, is analyzed in more detail. Other key players, such as HSF1 and immunophilins (peptidyl-prolyl isomerases), are also considered along with some partners of HSP90 and HSP70. HSP110 and HSP60 are here omitted because their roles as cancer stemness-promoting chaperones are not evident yet. 

### 3.1. HSP90

HSP90 belongs to the HSPC subfamily [41] and is an abundant cytosolic chaperone possessing ATP-ase activity and interacting with its client proteins in an ATP-dependent manner. Growth factor receptors, steroid hormone receptors, protein kinases, products of some oncogenes, and regulators of signal transduction and gene transcription are among the client proteins of HSP90 [50,51,57,62]. Many of those client proteins are regulators of cancer epigenetics and cancer cell signaling, which ensure invasive and metastatic growth of malignancies or their resistance to chemotherapy and radiotherapy [57,58,62]. Upon HSP90 dysfunction, these client proteins are inactivated, destabilized, and quickly degrade (the upper path in Figure 2), followed by the failing of the proliferative and defensive potential of tumor cells; consequently, it is expected that HSP90-targeting agents would sometime be used in cancer treatment (reviewed in [51]). 

#### 3.1.1. Intracellular HSP90 and Some of Its Partners in Chaperoning Client Proteins

Among the cancer-promoting HSP90 client proteins, there are surely some contributing to cancer cell stemness development or responsible for the manifestation of certain qualities of CSCs. For example, it is long known that telomerase (hTERT), a telomere end-restoring enzyme, which confers a capacity for endless mitotic division without replicative senescence [11]; survivin, an antiapoptotic protein contributing to the chemo- and radioresistance of CSCs [31]; HIF1α, a transcription factor involved in hypoxia-responsive signaling, which induces EMT and CSC niche formation and also chemo- and radioresistance of CSCs [2,3,4,5,17] MMP2, MMP9, and uPA, secretory proteinases assisting CSCs to migrate through the extracellular matrix and invade the vasculature for metastasis dissemination [16]; and EGFR, Akt, and Src, components of signal pathways resulting in the improved survival and higher motility of CSCs [2,3,4,5,17] are all HSP90 client proteins [50,57,62]. Lee et al. showed that the expression of HSP90α (HSPC2 [41]) was enhanced in human breast CSC-like cells enriched with ALDH while geldanamycin (an inhibitor of the HSP90 activity) was able to reduce the pool of such ALDH+ cells [63]. Later, it was revealed by clustered regularly interspaced short palindromic repeat (CRISPR)/Case9-mediated knocking out that HSP90α is required for the activities of tumor cells, such as migration, invasion, and metastasis spread [64], that are generally accepted attributes of CSCs. A recent study of Song et al. established a crucial role of HSP90A (HSPC1; [41]) in the stemness development in immune-refractory tumors: The NANOG-driven HSP90A/TCL1A/Akt pathway is responsible for the emergence of CSC-like tumor cells exhibiting an insusceptibility to immune attack, aggressiveness, and multi-modal resistance [27]. Mechanistically, HSP90A, being an NANOG transcriptional target, stabilized TCL1A by preventing its degradation at proteasomes; as a result, TCL1A mediated Akt activation and the acquisition of stemness-associated properties [27]. 

Using selective inhibitors of the HSP90 chaperone activity or techniques with HSP90 knockdown, it was shown that intracellular HSP90 can be conducive to EMT in carcinomas of different localization [65,66,67,68,69,70,71], while the mechanisms of such HSP90-dependent EMT may vary in different cases. So, intracellular HSP90 was reported to promote EMT via (i) activation of HIF1α and NFκB [65] and stabilization of an oncogenic nonhistone chromatin-binding protein high-mobility group AT-hook 2 (HMGA2) [66] in colorectal cancer cells, (ii) activation of the TGF-β1/Notch1 signaling [67] or anaplastic lymphoma kinase (ALK) signaling [68] in different variants of lung cancer, (iii) direct interactions with Twist1 to stabilize and activate the latter in hepatocellular carcinoma [69], (iv) stabilization of low-density lipoprotein receptor-related protein 5 (LRP5) followed by LRP5-mediated stimulation of Wnt/beta-catenin signaling in gastric cancer [70], and (v) activation of STAT3 followed by the Twist1 expression upregulation in ovarian, renal, and nasopharyngeal cancer [71]. 

Interestingly, inhibitory targeting of the HSP90 chaperone function may not only suppress EMT together with EMT-evoked CSC accumulation but also may reverse the cancer cell stemness by inducing MET in existing CSCs (both opportunities are designated in Figure 1). Subramanian and coauthors [72] reported that CSCs from head and neck squamous cell carcinoma decrease stemness markers as CD44, ALDH and vimentin but increase E-cadherin after treatments with two novel HSP90 activity inhibitors, Ku711 and Ku757. Such HSP90-inhibiting treatments resulted in depletion of the CSC fractions (CD44/ALDH double-positive cells) and downregulation of specific microRNAs associated with CSCs’ resistance to chemotherapy [72]. Taken together, all these publications [65,66,67,68,69,70,71,72] testify the important role of HSP90 in forming the CSC phenotype through EMT, although there may be multiple HSP90-mediated mechanisms of triggering and promoting EMT in different types of cancer.

Moreover, intracellular HSP90 appears to be necessary for the activation and functioning of some transcription factors and components of signaling pathways that maintain the stem phenotype in already formed CSCs or ensure manifestation of the cancer stemness-associated features. In particular, HSP90 is required for the functioning of HIF1α, which is critical for the stem phenotype’s maintenance in murine lymphoma CSCs [73]. Additionally, HSP90 promotes multicell spheroid formation, migration, and invasion of thyroid CSCs, and those CSC phenotype-related activities were sensitive to HSP90 inhibitors [74]. It was demonstrated on cultured glioma SCs that the association of HSP90 with EGFR/EGFRvIII preserves the receptor complexes from proteasomal degradation, thus supporting the work of cancer stemness-promoting pathways [75]. Under conditions of hypoxia and acidosis in the microenvironment of gliomas, tumorous (intracellular) HSP90 was upregulated in the hypoxic niches, which was correlated with both the CSC phenotype’s expression and HSP90-dependent HIF upregulation [76]. Other researchers described the HSP90-dependent expression of EGFR, EGFRvIII, and Akt in glioma SCs, which was required for their proliferation, migration, and survival, and also for angiogenesis [77]. In an in vitro model of starvation with depletion of glucose and serum factors in the esophageal adenocarcinoma cell line SKGT-4, the starvation-responsive increase in expression levels of HSP90 (and HSP70) was associated with the expression of CSC markers, such as CD44, ALDH1A1, and ABCG2 [78]. An increased HSP90α expression level was found in breast CSCs and this was associated with their self-renewal, owing to the upregulation of B lymphoma Mo-MLV insertion region 1 homolog (BMI1), whose expression in the mammary gland epithelium is oncogenic [79]. Such BMI1 expression in breast CSCs was due to the HSP90α-mediated activation of oncogene c-Myc product and enhancer of zeste 2 polycomb repressive complex 2 subunit (EZH2), which were both shown to directly interact with HSP90α, probably to facilitate their nuclear translocation and triggering of the BMI1 gene transcription; all these findings allowed the authors to suggest the HSP90α-dependent mechanism of the self-renewal of breast CSCs [79]. Similarly, a causal link between the HSP90 chaperone function, BMI1 expression, and CSC self-renewal was suggested for CSCs from head and neck squamous cell carcinoma [72]. However, another HSP90α-involving mechanism was proposed for CSC-like cells in triple-negative breast cancer: HSP90α together with GRP78 interacts with the zinc finger motif-containing C-terminal region of RPDM14, which was associated with cancer cell stemness, whereas the prevention of those HSP90α–RPDM14 interactions by HSP90 inhibitors or GRP78 inhibitors led to a decrease in the CSC-like cell populations (CD24−/CD44+ and SP cells) [80]. In a more recent publication dedicated to triple-negative breast cancer [81], the inhibitor (L80)-sensitive HSP90 chaperone function was shown to be necessary for activation of the Akt/MEK/ERK/JAK2/STAT3 signaling network, which generated CSC-like cell phenotypes (CD44+/CD24− cells with high ALDH1 activity, which are able to form mammospheres). An involvement of HSP90α in TRPM7-dependent HSP90α/uPA/MMP2 signaling was associated with extracellular proteinase activity, expression of cancer stemness markers, multicell spheroid-forming ability, and high metastatic potential of lung CSC-like cells [82]. In turn, HSP90β (HSPC3 [41]) was found to form complexes with clusterin, which seems to be critical for the HSP90 client protein levels, resistance to apoptosis, and spheroid-forming growth in gastric CSCs [83]. As the increased radioresistance is also one of the attributes of the CSC phenotype [2,3,4,5], HSP90 is thought to contribute to the radioprotective mechanisms acting in irradiated CSCs and spheroids [84,85]. Such a diversity of mechanisms and pathways involving intracellular HSP90 in the formation/maintenance of CSC phenotypes suggests an essential and multilevel contribution of this chaperone to the phenomenon of cancer-associated stemness.

The implication of intracellular HSP90 in the CSC phenotype’s regulation can, in turn, be modulated by some endogenous and exogenous factors. So, the HSP90 expression level in CSCs may depend on the HSF1 activity [55], TRPM7 expression [82], or signaling triggered by extracellular leptin [86]. The EMT- or cancer stemness-related HSP90 activities may be upregulated by histone deacetylase(s) via deacetylation of acetylated (inactive) HSP90 molecules, which leads to the chaperone function’s activation [67,87]. In gastric CSCs, the cancer stemness-related activities of HSP90 and the levels of its client proteins seem to be regulated by clusterin through its direct interactions with the chaperone [83]. An outcome of HSP90-mediated chaperoning can depend on the activity and expression level of CHIP (carboxyl terminus of HSP70-interacting protein), which functions as E3 ubiquitin ligase: Excess CHIP is able to switch the HSP90/HSP70 chaperone machine so that ubiquitination and proteasomal degradation of bound client proteins will prevail over their maturation and activation (see the upper path in Figure 2). Additionally, vice versa: The CHIP downregulation can preserve some proteins that otherwise undergo degradation. Both situations were modeled in human breast cancer [88,89]. YL-109-induced CHIP upregulation in MDA-MB-231 cells resulted in attenuation of CSC properties, including the mammosphere-forming ability, expression of stemness markers, invasiveness, and metastatic potential [88]. In a similar model, CHIP was found to be significantly downregulated in breast CSCs and such a CHIP deficiency was correlated with mammosphere formation in vitro, lung metastases in xenografted mice, and poor outcome among patients with breast cancer exhibiting low CHIP expression [89]. The authors explained such effects of the CHIP depletion by a failure in CHIP-mediated ubiquitination and degradation of OCT4, a transcription factor promoting stemness that is a client protein of HSP90; in support of that, they demonstrated direct CHIP–OCT4 interactions and also the CHIP depletion-mimicking action of a ubiquitination-defective OKT4 mutant [89].

Besides CHIP, other protein partners of HSP90 may somehow be involved in EMT and cancer stemness-associated events. For instance, the co-chaperone p23 was shown to contribute to the acquisition by prostate cancer cells of a more aggressive (CSC-resembling) phenotype with higher cell motility and pronounced capacity for invasion and metastasis formation [90]. There was a report that the co-chaperone Hop (HSP70/HSP90 organizing protein) and its complexes with cellular prion protein maintain the stemness in gliomas by ensuring proliferation and self-renewal in glioma SCs [91]. Meanwhile, both p23 and Hop are obligatory components of the HSP90/HSP70 chaperone machine (see Figure 2 and [49,50,51]).

The contribution of intracellular HSP90 to the development and maintenance of cancer stemness may be yet wider and deeper and include other HSP90-dependent mechanisms, for example, HSP90–telomerase (hTERT) interactions (it seems likely that they ensure endless mitoses of CSCs) or HSP90–immunophilin interactions (see [48,50], Figure 2, and further Section 3.6.). However, the suggested importance of those interactions for the CSC phenotype remains to be proven.

##### Targeting Intracellular HSP90

Cell-penetrating inhibitors of the HSP90 chaperone activity have been developed and studied for many years because this chaperone seems to be an extremely attractive target for antitumor therapy. Indeed, the functioning of HSP90 is known to be necessary for the activation of some oncogenes and cancer-related signaling pathways, which ensure malignant growth and tumor resistance to cytotoxic drugs and radiation exposure [50,51,57,62]. The dysfunction of cytosolic HSP90 in inhibitor-treated tumor cells may result in the inactivation and degradation of several cancer-promoting proteins, thus impairing the viability and adaptiveness of target tumor cells (see Figure 2 and [49,50,51]). Therefore, inhibitors of HSP90 chaperone activity may be used to repress tumors and sensitize them to therapeutics [51,57,58,62].

At present, several small molecule inhibitors of the HSP90 chaperone activity are at different stages of preclinical and clinical trials (see [51] for a review). In various models, some HSP90 activity inhibitors demonstrated quite hopeful results in suppressing EMT and/or repressing CSCs. So, experimental ways to prevent or reverse EMT have been described for inhibitors of HSP90 activity, such as NVP-AUY922 [66], ganetespib [65,68], KU711, and KU757 [72]. In many research groups, the CSC-repressing and/or CSC-sensitizing effects of the inhibition of intracellular HSP90 activity were observed with geldanamycin [63,76], AR-42 (a histone deacetylase inhibitor) [87], emodin [75], 17AAG [73,74,92,93,94], 17DMAG [79,80], NVP-AUY922 [92], WGA-TA [74], KU711 [72,74], Ku757 [72], L80 [81], and panaxynol [95]. In a system with in vitro and in vivo models, NVP-AUY922 has recently been used to inhibit the HSP90A-dependent TCL1A/Akt pathway, which confers stemness-like properties in immune-refractory tumors [27].

All those works provide only ‘proof-of-principle’ toward a possibility of the application of HSP90 activity inhibitors as CSC-targeting agents. So far, despite the huge efforts of researchers for a long time, no clinically applicable inhibitor of HSP90 activity has been developed yet. Besides routine problems with the toxicity and bioavailability of many HSP90-binding compounds, there may be another complication that is shown in Figure 5. 

There are opportunities to escape such a complication, as some inhibitors of HSP90 activity do not cause the HSF1 activation with subsequent expression of HSPs and MDR1. For example, KU711 [74] and panaxynol [95] were shown to inhibit HSP90 activity in CSCs without induction of HSP70 and gp170, which was correlated with better targeting of CSCs. Another relevant approach is to use HSP90 activity inhibitors in combination with other drugs, which somehow prevents the HSF1 activation/HSP induction pathway. So, treatments of CSCs with 17AAG in combination with the SIRT1 inhibitor amurensin G [92] or nonsteroidal anti-inflammatory drugs [94] exerted the desirable (CSC-repressing) effects without HSF1-mediated HSP70 induction and gp170 upregulation. Additionally, it seems promising to combine HSP90 activity inhibitors with inhibitors of the HSF1 activation or HSF1–HSE interaction (see Figure 3). The rationale of such an approach is supported by several studies performed on non-stem cancer cells: In model systems, co-treatments of tumor cell cultures with HSP90 activity inhibitors and known inhibitors of HSF1-mediated HSP induction, such as quercetin, KNK437, triptolide, and NZ28, significantly enhanced the cytotoxic and/or radiosensitizing effects thanks to prevention of HSF1-mediated HSP induction [96,97,98,99]. Similar results have been obtained for breast CSCs co-treated with geldanamycin and quercetin or KNK437 [63]. However, all these HSF1 inhibitors are not applicable in antitumor therapy and the problem remains to be resolved (see Section 3.5.).

It should also be said that there were attempts to target CSC-like cells through reducing the HSP90 expression in them. Booth et al. described that the GRP78 inhibitor OSU-03012 in combination with sildenafil markedly reduced the HSP90 level in glioblastoma-derived (CSC-like) GBM12 cells, growing as neurospheres, and killed them [100]. Gedunin, a plant-derived compound, was found to reduce the expression of HSP90 and its client proteins in human NTERA-2 CSC-like, which that was correlated with the enhancement of apoptosis [101]. These results seem quite encouraging, although it is still unknown how the HSP90 level-reducing inhibitory treatments affect HSF1; in any case, studies in this direction should be continued. 

#### 3.1.2. Extracellular HSP90

In turn, extracellular (secreted or cell surface bound) HSP90 can also play an important role in the development and manifestations of cancer cell stemness. At least, the implication of extracellular HSP90 in CSC phenotype regulation was clearly demonstrated for prostate cancer [102,103,104,105], breast cancer [106,107], and colorectal cancer [108]. 

It was shown for prostate cancer that extracellular HSP90 is conducive to EMT associated with the induction of high cell motility, invasiveness, and metastatic behavior [102]. Mechanistically, this mechanism was mediated by extracellular HSP90 via ERK signaling triggered by extracellular HSP90 and upregulation of matrix proteinases MMP2 and MMP9 when the enhanced extracellular HSP90 expression conferred augmented levels of MMP2/MMP9 mRNAs. In support of this, Nolan et al. reported a contribution of extracellular HSP90 to EMT and invasion in prostate cancer through stimulation of MEK/ERK signaling, leading to EZH2 transcription upregulation followed by the expression of Twist, Snail, and E-cadherin [103]. In another study, cytosolic HSP90 also recruited EZH2 to ensure the MBI1-mediated self-renewal of breast CSCs; however, this mechanism required direct HSP90–EZH2 interactions [79]. One can suggest the possibility that extracellular and intracellular HSP90 act in tandem to better engage EZH2-dependent pathways in the formation/maintenance of the CSC phenotype. In a later relevant publication [104], secreted HSP90 was characterized as a modulator of the CSC phenotype’s heterogeneity in prostate cancer: The extracellular HSP90 expression was correlated with upregulation of stemness-associated markers and the EMT effector Snail, while promoting both the self-renewal and spheroid growth and drug resistance in prostate CSCs. Interestingly, the in vitro formation of 3D (sphere-like) organoid structures by prostate tumor-derived cell lines was accompanied by the enhanced secretion of HSP90- and EpCAM-containing exosomes and also multiple stemness marker expression [105]; on the other hand, extracellular HSP90 appears to stimulate spheroid growth, EMT, and CSC accumulation. If an analogous scenario acts in vivo as well, this may be a certain type of tumor self-stimulation: Secreted HSP90 stimulates CSC phenotype formation while the newly formed CSCs secrete greater amounts of HSP90 and thus accelerate CSC generation. In addition, it is noteworthy that together with HSP70 and a set of EMT-associated biomolecules, HSP90 was found inside exosomes secreted by hypoxia-stressed prostate cancer cells [20]; such a presence of the chaperones among other exosomal components may be important for exosome formation and exosome-induced EMT/stemness, but this suggestion needs further examination. 

As for colorectal cancer, the levels of secreted HSP90α were shown to be correlated with enhanced cell migration, invasion, and metastasis development [108]. Such a contribution of secreted HSP90α to cancer stemness-related processes was dependent on and causally linked to transcription factor 12 (TCF12): Extracellular HSP90α triggers the CD91/IKK/NFκB signaling cascade, which results in TCF12 expression, while the latter is required for HSP90α secretion and EMT [108].

Studying breast cancer-derived cell lines, Stivarou et al. found that extracellular HSP90 was overexpressed in mammosphere-forming cell cultures, while the mammospheres were enriched with CSC-like CD44(+)/CD24(−/low) cells; the authors suggested that extracellular HSP90 represents a phenotypic marker of breast CSCs [106]. It was recently reported that the secretion of HSP90 by breast CSCs is required for cancer-related signaling pathways and breast tumor xenograft growth in mice [109]; inhibiting the HSP90 secretion and/or extracellular HSP90 functions was therefore suggested as a potential approach to CSC-based therapy of cancer. 

Speculating about the mechanisms by which extracellular HSP90 promotes cancer cell stemness, one can notice that all the cited publications describe certain modulations of cancer-related signaling that probably occur in response to interactions of extracellular HSP90 with the cancer cell’s surface. One of the revealed mechanisms is the upregulation of MMP2 and MMP9 expression [102]. However, it seems likely that secreted HSP90, being a molecular chaperone, is able to directly interact with the secreted MMPs and such extracellular chaperoning enhances the proteinase activity aimed at local destruction of the extracellular matrix/connective tissue, thereby facilitating tumor cell migration, CSC niche formation, invasion, and metastasis occurrence. Additionally, the extracellular HSP90-induced activation of secreted MMP2 and MMP9, if it exists, may yield enhanced proteolysis of some immunoreactive proteins at the CSC surface, thus making CSCs yet less visible by the immune system. These hypotheses are supported by the fact that both MMP2 and MMP9 are client proteins of HSP90.

Describing a contribution of HSP90 to the cancer stemness phenomenon, it is important to emphasize that this group of chaperones is indeed deeply involved in the formation/maintenance of the CSC phenotype. This HSP90-based regulation of the cancer stemness is multilevel and ubiquitous because it takes place in both intracellular and extracellular spaces and is coupled to many different CSC-related features, pathways, networks, interplays, and effectors. Therefore, HSP90 looks to be a very promising target for a therapeutic attack on CSCs and overcoming such CSC-associated challenges as tumor invasion, metastases, and resistance to therapeutics.

##### Targeting Extracellular HSP90

It follows from the above subsection that extracellular (plasma membrane-bound or secreted) HSP90 plays an important role in cancer stemness development/manifestations; therefore, some therapeutic approaches may be aimed at the presence of this chaperone at the CSC surface or at the machinery of HSP90 secretion from CSCs. As for the latter, the inhibition of HSP90 secretion from CSCs by means of multifunctional magnetic nanoparticles (MNPs) seems to be a very promising approach because this MNP-exerted effect was accompanied by sensitization of CSCs to thermotherapy and chemotherapy [107]. Gong et al. have shown that metformin, a drug against type 2 diabetes, can suppress such an EMT/cancer stemness-associated phenomenon as metastasis occurrence by inhibiting HSP90α secretion in a phosphorylation-dependent manner involving AMP-activated protein kinase α1 and protein kinase Cγ [109].

Moreover, the development and use of cell-impermeable HSP90 inhibitors deserve attention in respect to therapeutic targeting of CSCs. In a model with human and murine tumor cells, Tsutsumi et al. demonstrated that DMAG-N-oxide, a cell-impermeable inhibitor of HSP90 activity, suppresses cell migration, invasion, and metastasis formation in the lungs of mice [110]. Mechanistically, it was due to the inhibitor-induced disturbance of cell motility based on integrin/extracellular matrix-dependent rearrangement of the cytoskeleton, which required the involvement of extracellular HSP90 [110]. Although the authors did not operate the term “CSCs” in this publication [110], based on their results, one can suggest that CSCs, as mobile, invasive, and metastasis-forming tumor cells, were the major target of the inhibitor used. Besides, it seems likely that cell-impermeable inhibitors of HSP90 activity are able to downregulate cancer stemness-associated extracellular proteinases, such as MMP2 and MMP9, whose expression, secretion, and activity are dependent on extracellular HSP90 [102]. Here, one can notice the important advantages of cell-impermeable inhibitors of HSP90 activity: (i) They are much less toxic for normal cells than cell-penetrating HSP90 inhibitors and (ii) they do not activate HSF1, so the problem shown in Figure 5 will not arise in the case of the application of such inhibitors against tumors.

A very intriguing finding was reported by Crowe et al., who synthesized and used a cell-impermeable fluorophore-tagged HSP90 inhibitor [111]. This fluorescent probe was shown to bind to HSP90 expressed on the tumor cell surface in vivo and then to be internalized together with the chaperone. Such a phenomenon of inhibitor-provoked internalization of tumor cell surface-associated HSP90 was suggested as a potential way for the targeted delivery of drugs into aggressive malignant cells [111]. If such a way works toward CSCs as well, this will enable, first, the removal of HSP90 from the CSC surface (that is obviously harmful for CSCs) and, second, the introduction of something (e.g., a cytotoxic agent) into CSCs. It remains to be examined whether the same mechanism of HSP90 internalization acts in CSCs and is applicable for therapeutic targeting of them.

Finally, HSP90 expressed on the CSC surface may be a target for immunotherapy of cancer. As HSP90 was suggested to be a specific surface marker of human breast CSCs, an anti-HSP90 monoclonal antibody 4C5 was produced, which inhibited both the cancer stemness-related activity of human breast cancer cells in vitro and the primary growth of mammosphere-derived tumors in immunodeficient mice [106]. Monoclonal antibodies with a high specificity/affinity for CSC surface-bound HSP90 may be conjugated to certain toxins or radionuclides and thus be used in vivo as vectors for the targeted delivery of cell-killing agents to CSCs. 

### 3.2. HSP70

Members of the HSP70 (HSPA [41]) subfamily are major chaperones of eukaryotes. The constitutively expressed HSP70 catalyzes polypeptide chain folding and this chaperone function is performed in an ATP-dependent manner under synthesis, transport, or degradation of protein molecules [40,57,112]. A set of (co-)chaperones and co-factors regulates the interactions of HSP70 with ATP, ADP, and protein substrates; in cooperation with Hip, Hop, HSP40, HSP90, and CHIP, HSP70 works in the ATP-consuming chaperone machine, which determines the destiny (stabilization or degradation) of various client proteins (see [49,50,51] and Figure 2). Cytosolic HSP70 together with HSP90 controls the status of HSF1, preventing its activation under non-stressful conditions (see Figure 3). Besides, HSP70 cooperates with other intracellular HSPs, such as HSP110, HSP60, and HSP27, which can affect the processes of protein folding or protein degradation in the cytoplasm and organelles [112].

Under proteotoxic stresses, inducible HSP70 is the major protein product of the HSF1-mediated stress response that confers recovery and transient stress tolerance in HSP70-enriched cells (Figure 3). The HSP70-conferred cytoprotection is mainly based on the capability of excess HSP70 to attenuate the consequences of proteotoxic exposure and block apoptotic pathways in the stressed cell [40,57]. HSP70 is also able to protect normal and tumor cells from some drugs and radiation exposure [57,113].

The role of HSP70 in oncogenesis is of extreme importance and most human malignancies exhibit increased HSP70 expression; the latter is often correlated with tumor aggressiveness and poor tumor response to therapeutics [40,57,113]. Interestingly, cancer cells can expose HSP70 on their surface and secrete it out [57,114]. A contribution of intracellular and extracellular HSP70 to the cancer cell stemness is considered herein. 

#### 3.2.1. Intracellular HSP70

Enhanced HSP70 expression was found in CSC-like cells from human gastric cancer [115] and medulloblastoma [116]; in the latter case, a correlation between HSP70 and the NFκB complex was revealed. Using murine breast cancer models, Gong et al. found upregulated HSP70 in CSC-like tumor cells exhibiting both elevated levels of cell-surface stemness markers (CD44 and Sca1) and high metastatic potential [117]. In the same study, HSP70 knockout was shown to reduce a pool of tumorigenic cells with the CSC-like phenotype and impair the invasion and metastasis formation; at the molecular level, these effects of HSP70 gene inactivation were associated with reduced activation of the oncogenic c-Met protein [117]. In another model with HSP70-2 (HSPA1B [41]) gene silencing in human ovarian cancer cells, HSP70-2 knockdown upregulated the epithelial markers E-cadherin and cytokeratin, whereas it downregulated a set of EMT- and cancer stemness-associated proteins, including N-cadherin, vimentin, Snail, Slug, Twist, MMP2, MMP9, and others [118]. These HSP70 knockdown-induced alterations in the protein profile’s expression toward the non-stem cancer cell phenotype were accompanied by a shift towards the upregulation of known effectors of apoptosis, such as cytochrome-c, caspase 3, caspase 7, caspase 9, Apaf1, and others, and downregulation of pro-survival and anti-apoptotic proteins, including poly (ADP-ribose) polymerase 1 (PARP1), Bcl-2, Bcl-xL, survivin, XIAP, and others [118]. This is also good evidence of HSP70’s contribution to cancer cell stemness because resistance to apoptosis is known to be one of the characteristic signs of CSCs (see the subsection introduction). Overall, except one publication describing rather hypothetical mechanisms by which HSP70 might suppress CSC phenotype development [119], all other reports testified that intracellular HSP70 promotes cancer cell stemness.

Here, it is important to add that the contribution of intracellular HSP70 to the formation/maintenance of CSC-like phenotypes can be regulated by other protein factors. In particular, the binding of nestin with cyclin D1 and heat shock cognate protein HSC71 (HSPA8 [41], a non-inducible form of the cytosolic HSP70 chaperone) and also nestin-dependent phosphorylation of HSC71 along with the fact that nestin and HSC71 reciprocally regulate expression levels of each other were revealed in glioblastoma cells [120]. Those nestin–HSC71 interrelations seemed to maintain the CSC phenotype and such typical manifestations of the cancer stemness as tumorigenicity, invasion, and spheroid formation. Moreover, alterations in the CHIP expression/activity levels may affect the destiny of some stemness-regulating proteins bound to HSP70 and HSP90, as it was shown for breast CSCs [88,89].

It should be noted that HSP70 is the most inducible chaperone whose intracellular level can increase several-fold after HSF1-activating stressful treatments (see Figure 3). In vivo, HSP70 can be upregulated in normal and cancer cells in response to pathophysiological stress or therapeutic interventions. Although human malignancies usually exhibit enhanced HSP70 expression [40,57,113], the HSP70 level may transiently be even more increased in cancer cells stressed by hypoxia, starvation, and acidosis in tumor regions with poor vascularization. Taking into account that such hypoxic regions are known to be sites where EMT occurs and CSCs are generated [2,3,4,5], one can suggest that stress-induced HSP70 upregulation promotes the development and maintenance of cancer stemness. In support of this, Tawfeeq et al. observed correlations between stress-responsive enhancement of the expression of HSP70 (along with HSP90) and expression of cancer stemness markers CD44, ALDH1A1, and ABCG2 in esophageal adenocarcinoma cells (SKGT-4 line) undergoing starvation with the depletion of glucose and serum factors [78].

Likewise, HSP70 may be increased in cancer cells and CSCs as a result of drug administration or radiation exposure. For example, an increase in the HSP70 level was found in CSC-like cells (SP cells) isolated from breast cancer MCF-7 cells γ-irradiated at a dose of 5 Gy and this effect might be associated with the elevated radioresistance of the SP cells studied [121]. HSP70 also accumulated in breast CSC-like cells treated with geldanamycin, an HSP90 inhibitor [63]. As inducible HSP70 possesses a huge cytoprotective potential of a wide spectrum [40,57,112,113], the increased level of this chaperone in CSCs, if it takes place, may be one of the endogenous factors ensuring the high resistance of CSCs to hypoxia, oxidative or metabolic stress, apoptosis, replicative senescence, and therapeutics.

##### Targeting Intracellular HSP70

The previous subsection characterized intracellular HSP70 as one of the determinants of cancer stemness, so that its attractiveness as a target to attack CSCs is beyond doubt. In this respect, it seems possible to reduce the basal HSP70 level in CSCs by treating them with certain compounds. For instance, a decrease in intracellular HSP70 was observed in neurosphere-forming (CSC-like) glioblastoma GBM12 cells treated with a combination of the GRP78 inhibitor OSU-03012 and sildenafil, which was accompanied by post-treatment cell killing [100]. Monobenzyltin Schiff base complex, [N-(3,5-dichloro-2-oxidobenzylidene)-4-chlorobenzyhydrazidato] (o-methylbenzyl) aquatin (IV) chloride, C1 complex was demonstrated to reduce the HSP70 expression level in MCF-7-derived CSCs, which was accompanied by apoptosis induction [122]. In principle, these results confirm the possibility of targeting HSP70 in CSCs and suggest a therapeutic benefit from such targeting.

Besides suppressing the constitutive HSP70 expression in CSCs, it seems important to prevent undesirable HSP70 induction in response to treatments of CSCs with some HSF1-activating drugs or HSP90 inhibitors. In such a situation, Lee et al. successfully used quercetin and KNK437 to abolish HSF1 activation/HSP70 induction in breast CSCs whose HSP90 was inhibited by geldanamycin [63].

A number of small molecule inhibitors of HSP70 chaperone activity are known [56,57,113]; some of them exert antitumor effects and are considered and tested as potential tools in the fight against cancer. Importantly, pifithrin-µ (2-phenylethynesulfonamide), a known HSP70-inhibiting agent, was found to induce cell death (necroptosis) in malignant mesothelioma cells while stimulating EMT in the surviving cells and thus increasing the risk of metastases [123]. This finding suggests that not every inhibitor of chaperone activity is applicable against CSCs. Apparently, the preclinical trials of novel inhibitors of HSP70 (and of other chaperones) should include tests on EMT induction and the development of a pro-metastatic (CSC-like) phenotype. 

Here, it should be remembered that cytosolic HSP70 and HSP90 are endogenous suppressors of HSF1 (see Figure 3). Consequently, treatments of CSCs with inhibitors of the HSP70 chaperone function may cause the activation of HSF1 and induction of cytoprotective HSPs and gp170 followed by adaptation of the surviving cells, i.e., a scenario shown in Figure 5. If this happens, the approach of Lee et al. used for HSP90 inhibition in CSCs [63] may be effective for HSP70 inhibitors as well. In any event, the search for suitable CSC-targeting agents among small molecule inhibitors of HSP70 should be continued.

#### 3.2.2. Extracellular and Cell Surface-Bound HSP70

HSP70 and its complexes with peptides, being isolated from hepatocarcinoma tissue and added to hepatocellular carcinoma cells (Huh-7), promoted EMT to act as an external stimulus for the p38/MAPK signaling pathway [124]. Using a similar approach, Nigro et al. added exogenous recombinant HSP70 from *Arabidopsis thaliana* to breast cancer cells (MCF-7 and MDA-MB-231 cell lines) and observed an enhancement of cell migration, MMP activation, increased expression of survivin and cyclin D, and other phenotypic alterations toward a CSC-like phenotype with high cytoprotective and metastatic potentials [125]. Both those publications [124,125] describe artificially modeled situations, but it provides indirect evidence that extracellular (secreted) HSP70 can drive cancer cells to EMT- and CSC-like phenotypes. In support of this, there are data that cancer cells do secrete HSP70 [56,114].

Later, plasma membrane-bound HSP70 was proposed to be used as a specific easily detectable marker of CSC-like circulating tumor cells that have undergone EMT and consequently lost the epithelial cell surface markers EpCAM and CD326 [126]. As well as HSP90, HSP70 was found in exosomes secreted by prostate cancer cells undergoing hypoxic stress [20]. Although the biological significance of the latter finding is not yet clear, it seems likely that exosomal HSP70 somehow contributes to the CSC phenotype/niche formation. 

The above facts characterize extracellular HSP70 as a factor implicated in EMT induction and CSC phenotype development, while HSP70 expressed on the surface of circulating CSCs seems to be a unique target to attack these cells. Specific monoclonal antibodies recognizing HSP70 on the surface of CSCs may be one of the tools for such attacks aimed at the elimination or inactivation of CSCs; hypothetically, antibody targeting of HSP70 on the surface of CSCs may (i) promote their immunogenic cell death or (ii) inhibit their cancer-aggravating activities, or (iii) be used for the delivery of cell-killing agents to them. Another approach to targeting extracellular HSP70 in CSCs may be the creation of cell-impermeable inhibitors of HSP70 chaperone activity as it was made for extracellular HSP90 [110].

### 3.3. HSP40

HSP40 (the DnaJ subfamily [41]) is a partner of HSP70 in the ATP-dependent machinery of protein folding: HSP40 regulates HSP70 ATP-ase activity and ATP/ADP exchange, which is critical for interactions of HSP70 with protein substrates (see Figure 2 and [50,51]). HSP40 is thought to play an important role in cancer and the cancer stemness [61,127].

It was reported in 2012 that DnaJB8 promotes CSC phenotype development in renal cell carcinoma: Being overexpressed, DnaJB8 increased the percentage of CSC-like SP cells and enhanced their tumorigenicity, whereas the attenuation of DnaJB8 diminished the amounts of SP cells whose tumorigenicity became impaired [128]. Later, DnaJB8 overexpression in colon cancer cells was shown to enhance both the expression of stemness markers and tumorigenicity, thus confirming the contribution of this chaperone to the CSC phenotype’s formation [129]. Using DnaJB8 gene knockout in renal cell carcinoma, Yamashita et al. demonstrated diminished ratios of SP cells and the impaired spheroid-forming ability in DnaJB8-deprived renal cell carcinoma cells [130]. In the same study, DnaJB8 knockout in renal cell carcinoma cells conferred them sensitivity to docetaxel, thus indicating a link between HSP40 and drug resistance intrinsic to CSCs. Notably, an increase in the amounts of the SP cells and SOX2 expression was found in kidney cancer cells being subjected to heat stress; by means of DnaJB8 knockdown with siRNAs, it was shown that the observed effects were due to HSF1-induced DnaJB8 upregulation [131]. The revealed fact that both the accumulation of CSC-like cells (SP cells) and the expression of SOX2 (a transcription factor maintaining the self-renewal of CSCs [2,3,4,5]) are dependent on HSP40 and HSF1 is of importance. As the HSF1 activation and subsequent induction of HSPs (including HSP40) can be provoked by hypoxia and/or low pH, an analogous HSF1/HSP40-dependent mechanism may act in hypoxic tumor regions, thereby contributing to EMT and the emergence of CSCs.

The recent proteomic investigations identified another member of the HSP40 subfamily, DnaJB4, as a new driver of EMT in breast cancer [132]. The same researchers showed that the suppression of DnaJB4 in breast cancer cells with a mesenchymal phenotype impaired their migration capacity in vitro and reduced both primary tumor growth and lung metastasis occurrence in vivo.

The combination of the GRP78 inhibitor OSU-03012 and sildenafil was declared as a “universal” remedy to suppress in cancer cells (and CSCs) the expression of all major chaperones, including HSP40 [100]. In fact, the OSU-03012/sildenafil treatment was shown to reduce the level of HSP40 (as well as of the other chaperones) in glioblastoma-derived CSC-like GBM12 cells growing as neurospheres; the chaperone level decrease was associated with the cytotoxic effect [100].

Although KNK437, a benzylidene lactam compound, is known as an inhibitor of the HSF1-mediated expression of all inducible HSPs [133], Yang et al. revealed that in colorectal cancer, this inhibitor dramatically decreased the level of DnaJA1 (a member of the HSP40 family), while only slightly affecting the levels of other HSPs [134]. In this study, KNK437 reduced metastasis formation, which allowed the authors to suggest an important role of DnaJA1 in promoting the pro-metastatic (CSC-like) phenotype development.

The above data allows us to consider the HSP40 family members as key players in chaperone-involving pathways that lead to the formation/maintenance of the CSC phenotype. It seems that HSP40 is associated with the main features of cancer stemness, such as the expression of SC phenotype markers, tumorigenicity, self-renewal, migration, spheroid formation, drug resistance, and metastatic potential. Therefore, HSP40 seems to be a promising therapeutic target in the fight against CSCs. The described effects of KNK437 [134] and the OSU-03012/sildenafil combination [100] are the “proof-of-principle” that HSP40 can be significantly diminished by means of small molecule inhibitors. Therefore, such inhibitors have to be developed and tested in CSC-related models.

### 3.4. HSP27

This “small” ATP-independent chaperone (HSPB1 or referred to as HSP25 in rodents) has multifaceted functions in the mechanisms of cell regulation and cytoprotection. HSP27 can assist the protein refolding/degradation machinery and attenuate the aggregation of damaged protein molecules in the stressed cell [39,40,52,135]. Moreover, HSP27 is known to be an important component of cellular signaling: HSP27 undergoes phosphorylation as the terminal substrate in the p38/MAPK pathway; oligomeric structure and cytoprotective activities of HSP27 also depend on its phosphorylation status [135]. There is crosstalk between the p38/MAPK/HSP27 pathway and Akt signaling: phosphorylated HSP27 can induce phosphorylation (activation) of Akt [136]. Endogenous HSP27 can be a powerful suppressor of apoptosis and a cytoprotectant against oxidative stress [52,135].

It is generally accepted that HSP27 is implicated in oncogenesis and contributes to drug resistance and radioresistance of tumor cells [52,135,137]. First indications about a link between HSP27 and cancer stemness appeared in 2010: The small chaperone was shown to be involved in TGF-β1-induced EMT in lung cancer cells (A549 cell line) and this involvement was independent of Smad [138]. Then, the contribution of HSP27 to EMT was found in several different cancer-related studies and models, including breast cancer [139], gastric cancer [140,141], prostate cancer [142,143], renal cell carcinoma [144], and salivary adenoid cystic carcinoma [145]. These HSP27-associated mechanisms of EMT were similar in some cases of cancer while different in others. So, the significance of HSP27 for TGF-β1-induced EMT was demonstrated in four distinct types of carcinomas [138,141,144,145]. Notably, diverse HSP27-involving mechanisms were described for different inducers of EMT. For example, the HSP27-mediated upregulation of Snail and Prrx1 seemed to promote TGF-β1-induced EMT in salivary adenoid cystic carcinoma [145]. In the case of IL-6-induced EMT in prostate cancer, silencing HSP27 reversed the phenotypic transition by impairing MMP activity, cell migration, and invasion [142]. At the molecular level, it was due to the HSP27 depletion-associated decrease in IL-6-dependent phosphorylation of STAT3 and, as a consequence, a decrease in STAT3 binding to the Twist promoter [142]. However, in a model of EGF-induced EMT in prostate cancer cells, HSP27 was required for modulation of the EGF/Akt/β-catenin/Slug signaling pathway [143]. In this study, silencing Hsp27 decreased EGF-dependent phosphorylation of β-catenin on tyrosine 142 and 654, which, in turn, led to the enhancement of β-catenin ubiquitination and degradation, thereby inhibiting nuclear translocation of β-catenin and its binding to the Slug promoter.

There are also reports linking the p38/HSP27 pathway to EMT in colorectal cancer [146], oral cancer cells [147], renal cell carcinoma [144], and lung cancer [148]. Contrary to this, Fang et al. found that inactivation of p38 favors manifestations of CSC properties in non-small cell lung cancer cells because HSP27 phosphorylation mediates ubiquitination and proteasomal degradation of stemness-driving proteins, such as SOX2, OCT4, NANOG, KLF4, and c-Myc [149]. This means that in various types of cancer, phosphorylated HSP27 may play different roles right up to opposite ones.

Proteomics-based studies have identified HSP27 as a cancer stemness-associated protein [141,150]. Indeed, many publications describe various HSP27-dependent mechanisms ensuring certain manifestations of cancer stemness. In particular, HSP27 was reported to promote migration, invasion, and MMP activation in prostate cancer cells and glioblastoma cells undergoing EMT [143,151]. Increased HSP27 levels (along with the increase in HSP70) were revealed in radioresistant CSC-like SP cells isolated from a human breast cancer MCF-7 cell line [121]; a causal connection between HSP27 expression and the high radioresistance of CSCs was suggested by the researchers. Using HSP27 knockdown, Wei et al. demonstrated that HSP27 is required for the maintenance of tumorigenic ALDH+ breast CSCs exhibiting the ability to migrate and form mammospheres, and also for the expression of Snail and vimentin in them [139]. In this case, the contribution of HSP27 was realized through the HSP27-mediated activation of NF-κB by modulating the expression/activity of IκBα [139]. In a similar model with breast cancer, HSP27 was found to be responsible for vasculogenic mimicry activity in populations of mammosphere-forming CD24(−)CD44(+)ALDH(+) CSCs [152]; the activity’s manifestation was mediated by the EGF-triggered signaling pathway, resulting in HSP27 phosphorylation. By means of silencing HSP27 and its overexpression in salivary adenoid cystic carcinoma cell lines, it was shown that HSP27 increases CSC-like (CD133+/CD44+) cells exhibiting radioresistance, reduced E-cadherin levels, and the enhanced capacity for cell migration and invasion; all these effects were due to the HSP27-mediated upregulation of Snail1 and Prrx1 expression [145]. HSP27-associated resistance to apoptosis, oxidative stress, and some chemotherapeutic agents was reported for lung CSCs [148,153], oral CSC-like cells [147], colon CSCs [154], and esophageal CSCs [155]. Importantly, phosphorylated HSP27 seemed to confer resistance to caspase-dependent apoptosis induced by hypoxia or serum depletion in CD(133+) CSCs from colorectal, lung, brain, and oral cancer; this HSP27 phosphorylation-mediated adaptive mechanism was based on the p38 MAPK/MAPKAPK2/HSP27 pathway while being suppressed by protein phosphatase PP2A, which dephosphorylates HSP27 [146]. The same players, p38 MAPK, MAPKAPK2, HSP27, and PP2A, were found to be involved in an antiapoptotic mechanism ensuring the resistance of colorectal CSCs to anti-angiogenesis therapy [154]. Another variant of HSP27-dependent signaling, namely the HSP27/Akt/HK2 pathway, was shown to be characteristic of esophageal CSCs and responsible for their specific reprogramming of energy metabolism in favor of higher glycolysis and oxidative phosphorylation [155] (see also [38]). Taken together, these data [146,154,155] indicate that HSP27 participates in the adaptation of CSCs to microenvironmental stresses when a lack of oxygen, nutrients, and serum factors takes place in poorly vascularized regions of tumors, and the hypoxic CSC niche is formed.

Here, it should be emphasized that HSP27’s involvement in cancer stemness and EMT depends on the levels of its expression and functional activity. If HSP27 expression is mainly defined by HSF1 (and the HSF1 expression/activation/inactivation modulators), the activity of HSP27 is largely regulated via its phosphorylation/dephosphorylation, which is managed by MAPKAPK2, PP2A, and their own regulators. Interestingly, antisense long noncoding RNAs (lncRNAs) may also affect HSP27 phosphorylation/dephosphorylation, as BX357664 was found to downregulate the EMT-driving TGFβ1/p38/HSP27 pathway in renal cell carcinoma [144]. 

According to the data cited above, the “small” chaperone is really implicated in various EMT-promoting mechanisms and ensures manifestations of certain CSC phenotype-associated properties, such as high migration activity, invasiveness, metabolic reprogramming, expression of stemness biomarkers, resistance to apoptosis, radiation, drugs, hypoxia, and oxidative stress, etc. Consequently, HSP27 should be considered as a very attractive target for anti-CSC therapy, especially since there are at least several potential options to suppress its expression and/or activity. 

#### Targeting HSP27

In order to impair the contribution of HSP27 to cancer stemness, there were approaches to suppressing the expression of this chaperone in CSCs. Plant-derived compounds, such as quercetin [63,139] and ovatodiolide [156] and also a mushroom-derived methyl antcinate [157], were shown to decrease HSP27 expression levels in breast CSCs and exert CSC-repressing effects. It has to be elucidated whether these compounds are clinically applicable against breast cancer.

The “universal” combination of OSU-03012/sildenafil, being used against CSC-like glioblastoma GBM12 cells, reduced the HSP27 level and yielded cell killing [100]. Taking into account that the same combinative treatment also reduced the levels of all major HSPs and also GRP78 and GRP94, while killing cancer cells and CSC-like cells [100], one can wonder why this remarkable (in all respects) combination is so far not adopted for cancer treatment. 

Shiota et al. obtained encouraging results with OGX-427, an antisense construct for the blocking of HSP27 expression [142]. In this study, the EMT-promoting activity of HSP27 was established, while OGX-427 reduced metastasis formation in a murine model of prostate cancer and reduced the amounts of circulating tumor cells in patients with metastatic prostate cancer. OGX-427 is currently in phase I–II of combinative trials and this agent of ‘gene therapy’ may become an effective tool to oppose cancer stemness. 

The enhancement of HSP27 expression in breast CSCs treated with geldanamycin, an inhibitor of HSP90 activity, was found to impair the cytotoxicity of the inhibitory treatment [63]. The researchers used quercetin or KNK437 to prevent the treatment-induced overexpression of HSP27 and thus increase the CSC sensitivity to geldanamycin. Interestingly, besides its inhibitory effect on HSP27 expression [63], quercetin is able to impair HSP27 phosphorylation in CSCs by inhibiting the p38 MAPK/HSP27 signaling pathway, which is critical for CSCs’ resistance to cisplatin [147]. Based on these data [63,139,147], one can suggest that quercetin (a widely distributed bioflavonoid) is a suitable agent for targeting CSCs. However, as quercetin is poorly soluble, while its effectual concentrations (dozens of micromoles) are not clinically achievable, this compound is hardly applicable in cancer treatment. 

It was demonstrated that treatments with docetaxel micelles suppress HSP27 expression in CSCs of triple-negative breast cancer and confer their thermosensitization [158]. The proposed method (docetaxel treatment + mild hyperthermia) seems quite applicable in the clinical setting, so there is a need to test this combination in relevant clinical trials.

### 3.5. HSF1 and HSF1-Activating Exposure

HSF1 by itself is not a molecular chaperone, but this transcriptional factor initiates the stress-responsive expression of inducible HSPs in mammalian (normal and cancer) cells, thus dramatically increasing the intracellular level of chaperones [55,159]. In the unstressed cell, HSF1 is in complexes with HSP90 and HSP70, which prevents its activation. In the cell undergoing any proteotoxic stress (heating, low pH, hypoxia, energy starvation or others), both HSP90 and HSP70 are recruited by stress-damaged protein molecules and consequently liberate HSF1; the latter is activated, becoming phosphorylated and trimerized, and translocated to the nucleus where it binds to the HSE in the promoter regions of the HSP genes to trigger their transcription (Figure 3 and [55,159]). Certain protein kinases (including stress-kinases p38, JNK, and ERK1/2) and protein phosphatases regulate HSF1 activation/inactivation [55]. Importantly, cell-permeable inhibitors of the HSP90 chaperone function are able to activate HSF1 and thus stimulate HSP induction in inhibitor-treated cells (see [63,96,97,98,99]).

The substantial role of HSF1 in oncogenesis has been established [159] and targeting tumorous HSF1 is now considered a unique therapeutic opportunity (reviewed in [160]). Notably, HSF1-activating stressful conditions, such as hypoxia, energy starvation, acidosis, and inflammation/edema, are also known as stimuli for EMT induction and the formation of the CSC phenotype/CSC niche (see the introduction section); accordingly, one can suggest that HSF1 activation in tumor cells somehow contributes to cancer stemness. Indeed, it was demonstrated that HSF1 promotes TGFβ-induced EMT, tumorigenesis, and metastases in a murine breast cancer model [161]; the EMT-promoting mechanism was due to HSF1-mediated stimulation of the RAS/RAF/MEK/ERK1/2 signaling pathway. Additionally, HSF1 was found to be required for EMT in ovarian cancer (including a spheroid growth model) [162], hepatocellular carcinoma [163], and pancreatic cancer [164]. In the latter case, the abnormal HSF1 activation (phosphorylation) resulting in EMT induction in pancreatic cancer cells was correlated with the failure of AMPK activation in pancreatic tumors [164]. Interestingly, even if HSF1 activation occurred in cancer-associated fibroblasts (non-cancerous cells in the tumor microenvironment), it was accompanied by EMT induction in oral squamous cell carcinoma cells, which, as a result, enhanced their migration and invasiveness [165]. In any case, activated HSF1 is a mediator of EMT in various carcinomas.

In addition to its EMT-promoting activities, HSF1 seems to be in charge of manifestations of certain hallmarks of the CSC phenotype. The failure of HSF1 ubiquitination by the ubiquitin ligase F-box/WD repeat-containing protein 7 α (FBXW7α) caused HSF1 accumulation and subsequent enhancement of the invasive and metastatic potential in human melanoma cells [166]. By means of knockdown and overexpression of HSF1 in breast cancer cell lines, a direct contribution of this transcriptional factor to the CSC phenotype was demonstrated: The HSF1 expression level was positively correlated with the CSC phenotype frequency, stemness marker expression, and drug resistance [167]. HSF1 knockdown in sphere-forming human A172 glioblastoma CSCs resulted in failing of the spheroid-forming capacity, reduced expression of SOX2 (a marker of stemness), and downregulation of MMP2 activity; such an addiction of the glioblastoma CSC phenotype to HSF1 was somehow supported the Bcl-2-interacting cell death suppressor (BIS), as BIS depletion led to a decrease in the HSF1 protein level [168]. In a breast cancer model, it was shown how one of the key players in the CSC phenotype formation, β-catenin, is regulated by HSF1 in a phosphorylation-dependent manner: The activating phosphorylation of HSF1 at serine 326 led to HSF1-mediated involvement of the RNA-binding protein HuR (human antigen R), which controls β-catenin mRNA translation [169]. Later, the significance of HSF1 phosphorylation at serine 326 for breast CSCs was confirmed by Carpenter et al., who revealed that the Akt-HSF1 signaling axis is responsible for the metastatic potential and self-renewal of breast CSCs [170]. Similarly, the activating phosphorylation of HSF1 at serine 326 was shown to be critical for the maintenance of gynecologic CSCs, although the researchers explained the effect of the HSF1 activation-triggered induction of HSP27 [171]. However, not HSP27 but members of the HSP40 family, DNAJB8 [131] and DNAJA1 [134], were determined as the critical products of the heat-induced HSF1 activation/HSP expression pathway that conferred the cancer stemness-like properties. Despite some discrepancy (HSP27 vs. HSP40), HSF1 activation and subsequent expression of inducible chaperones (HSPs) in tumor cells clearly promote the CSC phenotype’s acquisition.

Moreover, the fact should not be neglected that HSF1 activation in malignant cells can also trigger MDR1 (ABCB1) gene transcription (see [29] and Figure 3), thus increasing their chemoresistance. As both the enhanced expression of membrane ABC transporters and elevated chemoresistance are the characteristic signs of CSCs (see the introduction section), it seems logical to consider the HSF1-mediated MDR1 expression as an additional contribution of HSF1 to cancer stemness development.

It should be noted here that the therapeutic arsenal of oncologists also includes hyperthermia and inhibitors of proteasomes (e.g., bortezomib); CSCs were shown to be very sensitive to either of these exposures [172,173] and co-treatments with hyperthermia or proteasome inhibitors are suggested for sensitizing CSCs to chemotherapy and radiotherapy. The sensitizing effects appear to be due to the accumulation of misfolded proteins in heated or proteasome inhibitor-treated CSCs because these excess misfolded proteins recruit cytosolic chaperones, which thus become unable to maintain the chaperone-dependent features of CSCs, such as chemoresistance and radioresistance. Meanwhile, the same cause, namely the accumulation of misfolded proteins, can result in the activation of HSF1 and expression of additional chaperones (HSPs), which can impair the cytotoxic and sensitizing action of hyperthermia or proteasome inhibitors on CSCs (see Figure 6). The HSF1 activation and expression of inducible HSP70 and HSP27 were observed in cancer cells treated with proteasome inhibitors [96]. It seems likely that the use of blockers of HSF1 activation/HSP induction, as it was done in models with CSCs [63] and non-stem cancer cells [96,97,98,99], would improve the beneficial effects of hyperthermia and proteasome inhibitors on CSCs.

It follows from the above data that the cancer stemness-related transcriptional activity of HSF1 is regulated via its phosphorylation/dephosphorylation; besides, proteins, such as FBXW7α [166] and BIS [168], can affect the relevant function and expression level of HSF1. Thus, there are several points for modulating HSF1’s contribution to the formation/maintenance of the CSC phenotype. In this respect, HSF1 really appears to be a universal target to attack the cancer stemness because inhibiting HSF1 activation in tumors would yield an opportunity to downregulate the entire spectrum of inducible HSPs, which promote EMT and the generation/maintenance of CSCs. 

#### Targeting HSF1

Several small molecule inhibitors of this stress-responsive transcription factor are known; first of all, these are quercetin and KNK437, which were discovered and characterized many years ago [133,174]. Both the compounds were used for blocking the HSF1 activation/HSP induction pathway in CSCs treated with geldanamycin, an HSP90 inhibitor; such a blockade sensitized CSCs to the HSP90-inhibiting treatment [63]. For a similar purpose, Moon et al. used nonsteroidal anti-inflammatory drugs (NSAIDs) for the sensitization of CD44 high (CSC-like) leukemia K562 cells to HSP90 inhibition with 17AAG [94]. The co-treatment with 17AAG and NSAIDs suppressed the HSF1-mediated HSP induction, thereby enhancing the cytotoxic effect toward CSC-like K562 cells [94].

In another study, KRIBB11, a small molecule inhibitor of HSF1, was shown to kill breast CSCs with synergistic cytotoxicity if combined with MK-2206, an inhibitor of Akt [170]. 

Obviously, there is a great need for the creation of clinically applicable inhibitors of HSF1 activation, which would enable the opposement of the cancer stemness development and sensitization of CSCs to inhibitors of HSP90 chaperone activity. Probably, co-treatments with HSF1 activation inhibitors would help to improve the scenarios shown in Figure 5 and Figure 6. However, such inhibitors should be used with great care, as they may increase the sensitivity of the patient’s tissues and organs to chemotherapy and pathophysiological stresses, such as ischemia, inflammation, etc.

### 3.6. Immunophilins and Immunophilin-Like Peptidyl-Prolyl Isomerases

This family of molecular chaperones comprises peptidyl-prolyl isomerases, which regulate protein folding by catalyzing the *cis/trans* transitions at the pyrrolidine heterocycle of L-prolines integrated into a peptide bond [48,175]. Immunophilins can assist the HSP70/HSP90 chaperone machine in client protein folding (see the lower path in Figure 2 and [48,49,50,175]). Immunophilins are traditionally divided into two subfamilies according to their diverse capacity to bind certain immunosuppressive drugs: (i) FK506-binding proteins (FKBPs and FK506-binding protein-like or FKBPL) and (ii) cyclosporin-binding cyclophilins. Most immunophilins are in the cytosol [48,175], but cyclophilin D is localized to the mitochondrial matrix and performs very specific functions of controlling the mitochondrial permeability transition pore and regulating electron transport chain behavior [176]. The other peptidyl-prolyl isomerase, protein never in mitosis gene A interacting-1 (PIN1), is an immunophilin-like chaperone protein, which isomerizes phospho-serine/threonine-proline motifs only [177]; this is an example of site-directed chaperoning, which is dependent on the site-specific phosphorylation of a protein substrate.

Functioning as chaperones, peptidyl-prolyl isomerases control protein conformations and protein trafficking, steroid receptor action, activities of the transcription factor NFκB and hTERT telomerase, Akt/mTOR signaling cascade etc; due to their chaperone function, peptidyl-prolil isomerases are involved in the regulation of the cytoskeleton, apoptosis, cell cycle, and cell differentiation [175]. Meanwhile, FKBP51, FKBP52, and PIN1 can play an important role in onocogenesis and tumor resistance to therapeutics [48,175,177]. In contrast, FKBPL has been characterized as an antitumor protein; its peptide derivatives (AD-01 and ALM201) were taken for preclinical and clinical trials and demonstrated quite encouraging results in CSC-related models [178,179,180]. 

There are published data indicating a link between peptidyl-prolyl isomerases and cancer stemness. It was shown that FKBP12 knockdown increased the mammosphere formation and tumorigenicity in breast cancer cell lines [181]. By means of proteomics- and microarray-based investigations, FKBP4 was identified as one of the upregulated (marker) proteins in neoplastic SCs from childhood germ cell tumors [182], mammosphere-derived breast CSCs [150], and oral squamous cell carcinomas [183]. After gene expression analysis of CSCs isolated from various carcinomas and sarcomas, the peptidyl-prolyl isomerase A (cyclophilin A)-encoding gene was found among the four most stably expressed vital genes, which were proposed as ‘housekeeping genes’ for gene expression profiling aimed at comparison between the CSC phenotype and other cell phenotypes [184].

Importantly, peptidyl-prolyl isomerases are involved in the regulation of EMT in tumor cells. PIN1 was shown to be overexpressed in tamoxifen-resistant breast cancer MCF-7 cells, which have undergone EMT, while PIN1-siRNA treatments downregulated the expression of mesenchymal markers and Snail [185]. The same researchers suggested the involvement of PIN1 in EMT via PIN1-mediated stimulation of the PTEN/PI3K/Akt/GSK3β pathway and/or GSK3β/NFκB-dependent Snail activation. Then, Luo et al. [186] reported that PIN1 overexpressed in breast cancer cells can trigger EMT, while miR200c is able to overcome this effect. A PIN1-dependent mechanism of EMT was also suggested for EGFR-mutant lung adenocarcinoma cells resistant to tyrosine kinase inhibitors [187]. In a more recent study, PIN1 expression in human gallbladder cancer was found to be correlated with the activating phosphorylation of STAT3 (serine 727) and NFκB-p65 (serine 276), which induced EMT in tumor cells; accordingly, PIN1 knockdown impaired the phosphorylation/activation of both STAT3 and NFκB, which was accompanied by decreased Snail and ZEB2 expression in PIN1-depleted tumor cells, which lost a mesenchymal phenotype [188]. Using shRNA-mediated or pharmaceutical inhibition of PIN1, Zhang et al. showed that in gastric cancer, PIN1 promotes EMT- and CSC-related activities, such as cell migration, invasion, and lung metastasis occurrence [189]. In turn, FKBP51 has been characterized as one of the key players in promoting EMT in melanomas [190,191] and is also a promoter of NFκB-dependent EMT in papillary thyroid carcinoma [192]. A cyclophilin D activity-regulating mechanism was reported to be involved in EMT induced by p53Ψ, a transcriptionally inactive p53 isoform, which was suggested to trigger a pro-metastatic cellular program [193].

Besides their contribution to EMT in tumors, peptidyl-prolyl isomerases mediate the most characteristic properties and signs of the CSC phenotype. PIN1’s contribution to cancer stemness development seems especially significant and multifaceted. In particular, PIN1 was demonstrated to maintain breast CSCs by preserving Notch1 and Notch4 from proteasomal degradation mediated by the ubiquitin ligase Fbxw7α; such activity of PIN1 was conducive to Notch signaling in breast CSCs, which conferred their self-renewal and metastatic spread [194]. In another breast cancer-related study, PIN1 was characterized as one of the major drivers of breast CSCs and their self-renewal and tumorigenicity [186]. Notably, PIN1 appears to act downstream of miR200c, so that miR200c may suppress the breast cancer stemness-promoting activity of PIN1, while PIN1 overexpression may overcome the inhibitory effect of miR200c [186]. Later, the same research group reported further unraveling of the PIN1-dependent mechanism driving breast CSCs: Overexpressed PIN1 upregulates the transcription of genes encoding Rab2A (a small GTPase), and then excess Rab2A interacts with ERK1/2 to prevent the inactivating dephosphorylation of ERK1/2, which remains active and thus mediates ZEB1 upregulation and nuclear translocation of β-catenin [195]. According to Chen et al. [196], upregulated PIN1 was associated with cell migration, invasion, and metastasis occurrence in the case of pancreatic ductal carcinoma. In fact, PIN1 overexpression may be an essential cause for the tumorigenesis and cancer stemness development in breast and pancreatic carcinomas [186,195,196]. Studies of the roles of SPOP and PIN1 in prostate cancer stemness and progression have revealed that PIN1 is an upstream regulator of NANOG and also protects the latter from SPOP-mediated polyubiquitination and degradation, thus promoting NANOG-conferred cancer stemness features in prostate tumors [197].

The other member of the immunophilin family, FKBP51, was shown to be associated with the invasiveness and metastatic potential of melanoma SCs [190,191] and with the enhanced cell migration and invasion in papillary thyroid carcinomas [192], and also with cancer stemness, tumor recurrence, and poor prognosis in oral squamous cell carcinomas [183]. A contribution of FKBP4 to cancer stemness manifestations, such as self-renewal, spheroid formation, and chemoresistance, has been reported as well [181,182]. Meanwhile, a pivotal role of cyclophilin A in maintaining stemness in gliomas has been established: By means of direct binding to β-catenin, cyclophilin A regulates the interactions of β-catenin with Wnt target gene promoters and TCF4, thus resulting in an enhancement of transcriptional activity [198]. It follows from the latter publication that cyclophilin A promotes the self-renewal and radioresistance of neurosphere-forming (CSC-like) glioma cells through stimulating the Wnt/β-catenin pathway [198]. Even the intramitochondrial peptidyl-prolyl isomerase cyclophilin D may be involved in activating the motility, invasive, and pro-metastatic properties of CSCs, as was suggested by Senturk et al. [193].

Taking into consideration the important role of immunophilins in cell regulation [175,176,177], one can expect a greater contribution of immunophilins to the development and maintenance of the CSC phenotype. For example, the binding of FKBP51 and FKBP52 to HSP90, which is so important for the HSP90-dependent mechanisms of cell regulation [48,175], may somehow be related to the formation of the CSC phenotype as well. It seems likely that the HSP90-mediated involvement of transcriptional factors and signaling pathways in EMT induction or certain manifestations of cancer stemness (see Section 3.1.1.) requires direct HSP90–immunophilin interactions. In turn, the formation of heterocomplexes between FKBP51, FKBP52, HSP90, and telomerase (hTERT) [175] may be necessary for the endless division and apoptosis-free existence of CSCs. Nevertheless, the suggested significance of these HSP90–immunophilin interactions for cancer stemness remains to be established.

Except FKBPL [178,179] and FKBP12 [181], peptidyl-prolyl isomerases are ones of the endogenous drivers of cancer stemness, which control different points in tumor cell regulation, including gene transcription, signaling networks, the mitochondrial permeability transition pore, and others. Consequently, the cancer stemness-related activities of peptidyl-prolyl isomerases can be suppressed by specific inhibitors or certain microRNAs. Therefore, immunophilins and PIN1 are promising therapeutic targets for cancer treatment.

Intriguingly, endogenous FKBPL and FKBP12 exhibit anti-CSC activities. FKBPL knockdown in breast cancer cells upregulated the stemness markers NANOG, OCT4, and SOX2 and increased the CSC fraction [178]. In the latter study, FKBPL overexpression in breast cancer cells was shown to reduce the number of CSCs via downregulation of DLL4 and Notch4 [179]. In turn, FKBP12 knockdown in breast cancer cell enhanced the mammosphere formation and other stemness features, whereas FKBP12 overexpression exerted the opposite effects [181]. 

#### Targeting Peptidyl-Prolyl Isomerases and Use of FKBPL-Derived Peptides

The search for CSC-targeting inhibitors of peptidyl-prolyl isomerases currently continues. Two plant-derived compounds, curcumin and emodin, were demonstrated to suppress PIN1 expression in human cervical cancer cells via inhibition of the TGF-β-stimulated Wnt/β-catenin signaling pathway, which was accompanied by the downregulation of EMT markers, such as Snail and Slug [199]. Later, it was found that another plant-derived compound, celastrol, strongly suppresses PIN1 expression in ovarian cancer cells and this is accompanied by downregulation of the expression of CD44, NANOG, KLF4, and OCT4 as well as a decrease in the CD44 high/CD24 low cell population (i.e., CSC-like cells) [200].

A covalent PIN1 inhibitor, KPT-6566, was identified that selectively inactivates PIN1 by covalently binding to its catalytic site, which ends in PIN1 degradation [201]. Importantly, KPT-6566 treatments caused selective killing of malignant cells and also suppressed the CSC-like phenotype in vitro and metastasis growth in the lungs [201]. Another small molecule inhibitor of PIN1, PiB, inhibited both SPOP-mediated NANOG degradation and stemness-associated spheroid formation in prostate cancer cells [197]; the application of PIN1 inhibitors against prostate cancer with wild-type SPOP was suggested by the researchers. As for pharmaceutical targeting, all-trans retinoic acid-mediated inhibition of PIN1 was shown to eliminate gastric CSCs and also impair their self-renewal and tumorigenic potential [189]. 

Finally, peptidyl-prolyl isomerases may be targeted by means of ‘gene therapy’. In particular, Saw et al. constructed an aptamer-like peptide (aptide)-decorated liposomal nanoplatform for the targeted delivery of cyclophilin A siRNA to fibronectin-overexpressing glioma cells [202]. It was discussed that such a liposomal nanoplatform-based approach may be exploited in vivo for silencing cyclophilin A expression in glioma cells to overcome their stemness [202]. Another very original approach has been reported by Zhang and Zhang [203], who used microRNA (mja-miR-35) isolated from a shrimp, *Marsupenaeus japonicus*, to silence the PIN1 gene in human CSCs. As a result of the action of mja-miR-35, PIN silencing was selectively achieved in melanoma CSCs and breast CSCs, which was accompanied by impairment of their spheroid-forming ability and stimulation of their apoptotic death [203]. This implies that microRNAs from aquatic animals may successfully be used to oppose stemness in human malignancies.

Thus, as drivers of cancer stemness development, peptidyl-prolyl isomerases seem quite targetable, and after trials, their clinically applicable inhibitors have to be used for anti-CSC therapy. 

Notably, the situation with endogenous FKBPL shows how a molecular chaperone may provide a tool for effective targeting of CSCs. McClements et al. reported that treatments of breast cancer cells (MCF-7, MDA-MB-231, ZR-75 cell lines) with an FKBPL peptide derivative, AD-01, reduce the populations of CD44-positive CSCs and their ability to form mammospheres [178]. It was described that the AD-01 treatments suppressed both the self-renewal capacity of breast CSCs in vitro and tumor initiation in vivo, while the stemness markers NANOG, OCT4, and COX2 were reduced in the treated cells [178]. In the next work of the same research group [179], FKBPL-derived peptides (AD-01, preclinical peptide/ALM201, clinical peptide) were successfully used to reduce the numbers of CSCs and suppress the metastatic ability in MCF-7 and MDA-MB-231 breast cancer lines; the achieved effects were accompanied by significant impairment of cancer cell migration/invasion and downregulation of DLL4 and Notch4. According to the recently published data [180], the FKBPL-derived peptide, ALM201, reduced the numbers of CSCs in ovarian cancer cell lines and xenografts by targeting the CD44/STAT3 pathway and inhibiting angiogenesis. Here, it should be optimistically mentioned that AML201 has completed a Phase 1a clinical trials in ovarian cancer patients and other advanced malignancies [180]. 

## 4. Contribution of GRPs, TRAP1, Protein Disulfide Isomerases, and Calreticulin to the Regulation of the CSC Phenotype, and also Approaches to Overcoming Their Cancer Stemness-Promoting Activities 

This subsection is dedicated to molecular chaperones, which primarily reside inside organelles and whose expression and functions are associated with ER stress. GRPs, TRAP1, protein disulfide isomerases, and calreticulin are mainly localized to the ER and/or mitochondria; however, some amount of GRP94, GRP78, protein disulfide isomerases, and calreticulin can be exposed on the cell surface or secreted. Within the organelle compartments, these chaperones catalyze the (re)folding of proteins and assembly of protein oligomers or assist the degradation of aberrant protein molecules; such activities allow these chaperones to participate in the cellular stress response and also regulate the maturation/sorting/processing/export of (glyco)proteins, redox and Ca^2+^ balances, energy metabolism, apoptosis, phagocytosis, etc. [42,43,44,45,46,47,48]. GRPs, TRAP1, protein disulfide isomerases, and calreticulin are involved in oncogenesis and promote tumor resistance to therapeutics [42,43,44,45,46,47,48]. The cancer stemness seems to be dependent on these chaperones as well. GRP170 and calnexin are not mentioned here, as their roles in cancer stemness development/maintenance are not yet elucidated.

### 4.1. GRP94

This chaperone (often referred to as gp96 or HSPC4 [41]) plays an important role in oncogenesis; its main functions are thought to be the maintenance of Ca^2+^ homeostasis, protection against apoptosis, and chaperoning integrins, Toll-like receptors, and LRP6 to ensure their cell surface expression [204]. The specific (structurally determined) binding of GRP94 to the α7 helix region of the αI domain of integrins seems critical for cancer cell migration, invasion, and metastatic spread [205]. The enhanced expression of GRP94 was found in breast CSCs [206,207] and CSC-like cells derived from oral malignancies induced by the areca nut [208].

With the use of RNA interference (RNAi)-based silencing, it was shown that GRP94 is conducive to the aggressiveness of gliomas by activating such cancer stemness-associated properties as cell migration and invasion [209]; mechanistically, it was due to GRP94-mediated stimulation of the Wnt/β-catenin signaling pathway, one of the pivotal pathways in CSC phenotype formation. 

Taken together, these findings allow us to attribute GRP94 to the cohort of cancer stemness-promoting chaperones and suggest a therapeutic benefit of targeting GRP94. The “universal” combination of OSU-03012 and sildenafil was demonstrated to decrease the GRP94 level in CSC-like glioblastoma GMB12 cells along with decreasing the levels of GRP78 and other chaperones [100]. Meanwhile, specific small molecule inhibitors of GRP94 are currently being developed [204,210] and they need to be tested in CSC-related models.

### 4.2. GRP78

Being localized to the ER, GRP78 (sometimes referred to as BiP or HSPA5 [41]) is the main cellular sensor of ER stress; this chaperone triggers the UPR via liberating the three ER stress transducers (see Figure 4 and [43]). The functioning of GRP78 in the stressed cell is multifaceted and may yield opposite outcomes. Indeed, among the GRP78-dependent responses to ER stress are both cytoprotective events (induction of chaperones, chaperone-mediated refolding or degradation of stress-damaged proteins, prevention of apoptosis) and proapoptotic events (CHOP expression and activation of caspases) aimed at either cell survival or cell death, respectively [43,46]. By such a way, the ‘quality control’ program is realized: In the case of too severe damage, the prolonged UPR eliminates the stressed cell via apoptosis.

However, many facts indicate that GRP78 is the cancer-promoting chaperone that is involved in oncogenesis and drives unlimited malignant growth and tumor resistance to therapeutics [43,211]. Although GRP78 majorly resides in the ER, this chaperone is partly present in the cytoplasm, nucleus, mitochondrion, and at the cell surface or secreted into the extracellular space [43]. Both intracellular and extracellular GRP78 contributes to cancer stemness development/manifestations. 

#### 4.2.1. Intracellular GRP78

There were reports about the high intracellular GRP78 levels in CSC-like cells of different origins [206,207,208,212]. In 2010, Wu et al. demonstrated that knockdown of GRP78 in tumor-initiating (CSC-like) cells from head and neck cancer impairs their self-renewing capacity, tumorigenic potential, and expression of stemness-related genes, while reducing the SP cell’s proportion and enhancing apoptosis [213]. So, a causal connection between GRP78 expression and the CSC phenotype was revealed many years ago.

Many publications confirm the role of intracellular GRP78 as one of the EMT regulators in various human carcinomas [214,215,216,217,218,219,220,221,222,223,224]. It was shown that GRP78 promotes EMT in head and neck cancer cells [214], colon cancer cells [215], lung cancer cells [216,217,218,219], breast cancer cells [220,221], nasopharyngeal carcinoma cells [222], prostate cancer, and multiple myeloma cells [223]. EMT markers were found to be correlated with increased GRP78 in samples from patients with lung adenocarcinoma [224]. Contrary to all those publications [214,215,216,217,218,219,220,221,222,223,224], there are at least three research papers dedicated to hepatocellular carcinoma cells [225,226,227] in which EMT and cancer stemness were associated with downregulation of intracellular GRP78, which seems rather unexpected and hard to explain. Such intriguing findings force the suggestion of some unusual activities of GRP78 in hepatocellular carcinomas. This issue remains to be clarified, especially since the co-expression of GRP78 and markers of stemness was also observed in hepatocellular carcinomas [228].

In the cases when intracellular GRP78 promoted EMT in tumor cells, different GRP78-involving mechanisms were described. For example, GRP78 is required for several Cripto-1-mediated signaling pathways that induce EMT and increase the pool of CSCs [7]. In one study performed on colon cancer [215], the GRP78-activated NRF-2/HO-1 signaling pathway was suggested to promote EMT with decreased E-cadherin and increased vimentin expression in the tumor cells. However, in another similar study, overexpressed GRP78 was shown to stimulate EMT in colon cancer cells via an autocrine mechanism, enhancing the expression/secretion of TGF-β1 and triggering TGF-β/Smad2/3 signaling [229]. The enhanced expression of GRP78, Src, MAPK, and Smad2/3 seemed to be associated with hypoxia-induced EMT in lung cancer A549 cells [218]. More recent investigations on the same model have revealed the role of intracellular GRP78 as the key endogenous regulator of EMT in hypoxia-stressed A549 cells: Hypoxia upregulates the expression of GRP78 while increased GRP78, in turn, mediates the activating phosphorylation of Smad2/3, Src, p38, ERK, and JNK, thus promoting EMT via activation of the Smad2/3 and Src/MAPK signaling pathways [219]. In a model of non-small lung cancer, the GRP78-dependent mechanism of EMT was realized via the PI3K/Akt/Mdm2 signaling pathway, which was sensitive to GRP78 downregulation by the differentially expressed in adenocarcinoma of the lung (DAL-1) [217]. The importance of GRP78’s interaction with LOXL2 (lysyl oxidase-like 2) was established for EMT induction in breast cancer cells [220]; this interaction results in stimulation of the IRE1/XBP1 signaling pathway followed by expression of SNAI1, SNAI2, ZEB2, and TCF3 being the EMT drivers. Such a variety of GRP78-mediated mechanisms of EMT emphasizes the multifunctional role of intracellular GRP78 in cancer stemness regulation.

There are a number of publications indicating that intracellular GRP78 ensures certain properties of the CSC phenotype. Stable complexes between GRP78 and KIAA1199, being formed in the ER of breast cancer MDA-MB-435 cells, were shown to be necessary for KIAA1199-mediated maintenance of the mesenchymal status of cancer cells along with their enhanced capacity for migration and metastasis dissemination [230]. Approaches with the knockdown of GRP78 helped to elucidate that this chaperone is required for the self-renewal ability and radioresistance in CSC-like cell fractions from breast cancer MCF-7 cells [231] and also for chemo- and radioresistance in CD24(−)/CD44(+) stem cells of head and neck cancer [214]. By means of the selective inhibition of GRP78 ATPase with isoliquiritigenin (a chalcone-type bioflavonoid), it was shown that the binding of GRP78 to β-catenin is necessary to realize the GRP78/β-catenin/ABCG2 pathway in breast CSCs, which contributes to their drug resistance [232]. Later, GRP78-downregulating treatments with isoliquiritigenin enabled Hu et al. [233] to discover the essential role of GRP78 in the expression of CD44, ALDH1, and ABCG2 (all are stemness markers) and the ability for invasion and metastatic spread and resistance to chemotherapy in CSCs from oral squamous cell carcinomas. Silencing of GRP78 suppressed self-renewal and radioresistance of glioma CSCs, which was due to the increase in the microRNA-205 level [234]. In human pancreatic CSC-like cells treated with gemcitabine, GRP78-mediated upregulation of uPA enhanced invasion, spheroid formation, and resistance to apoptosis [25]. The significance of direct interactions of GRP78 (and HSP90α) with PRDM14 for the maintenance of CSC-like populations of CD24(−)/CD44(+) cells and drug-resistant SP cells has been demonstrated in a model of triple-negative breast cancer [80]. Additionally, in head and neck squamous cell carcinoma, the GRP78/p-PEK/NRF2 signaling pathway was found to promote the Warburg effect (domination of glycolysis over oxidative phosphorylation), being critical for the maintenance of both low ROS levels and the CSC-like phenotype in tumor-initiating cells [235]. Studies of the effects of shRNA/GRP78 on pancreatic cancer cells have revealed that GRP78, as the ER stress sensor and redox regulator, promotes self-renewal and tumorigenicity while preventing oxidative stress and thus maintaining cancer stemness [236].

Taken together, these published data indicate the multifaceted role of GRP78 in the formation and maintenance of the CSC phenotype. Despite the somewhat contradictory situation with hepatocellular carcinoma [225,226,227,228], it was shown in many relevant models that GRP78 promotes EMT and is responsible for the self-renewal and tumorigenicity of CSCs; their ability to invade and to form spheroids and metastases; their resistance to apoptosis, oxidative stress, chemotherapy, and radiotherapy; their energy metabolism switching; expression of stemness markers, etc. Meanwhile, the EMT- and cancer stemness-promoting activities of intracellular GRP78 may be downregulated by certain endogenous protein factors, such as DAL-1 [217], P4HB [227], and YAP1 [228], and also by a set of three microRNAs: miR495 [222], miR205 [234] or miR-30d, miR181a, and miR-199a-5p [237]. No doubt, tumorous GRP78 is one of the most promising targets to oppose cancer stemness.

##### Targeting Intracellular GRP78

Approaches to the suppression of EMT in cancer cells by means of inhibiting intracellular GRP78 were reported [216,221]. So, Yiqi Chutan Recipe (YCR, a Chinese herbal prescription) was found to inhibit or even reverse hypoxia-stimulated EMT in lung cancer A549 cells through the inhibition of GRP78 expression along with downregulation of Src, MAPK, and Smad2/3 [216]. Nayak et al. reported that IKM5 (2-(1-(1H-indol-3-yl)octyl)-3-hydroxy-6-(hydroxymethyl)-4H-pyran-4-one), a novel indolylkojyl methane analogue, inhibits EMT and invasion in breast cancer cells by binding to GRP78 and decreasing its expression along with the downregulation of EMT markers, such as vimentin, Twist1, and MMP2 [221]. The revealed ability of IKM5 to suppress the growth of lung metastases in a murine model allowed the researchers to suggest clinical development of this GRP78 inhibitor for the fight against breast cancer [221]. 

Additionally, approaches to targeting intracellular GRP78 with various natural and synthetic compounds have been studied in an attempt to oppose cancer stemness. Here, it is impossible not to again cite Booth et al., who used the GRP78 inhibitor OSU-03012 (AR-12) in combination with sildenafil to treat cultured brain CSC-like cells [100]. Such an in vitro treatment reduced the intracellular GRP78 level in neurosphere-forming glioblastoma GBM12 cells and killed them. As the same (combinative) treatment also reduced the intracellular levels of HSP90, HSP70, HSP60, HSP40, HSP27, GRP94, and GRP58 [100], it is hard to assert that the observed cytotoxic effect is due to the decrease in GRP78. Nevertheless, this method, which enables the simultaneous downregulation of all major chaperones in brain CSCs, deserves close attention, especially since each of the drugs is approved for use.

In another study dedicated to targeting GRP78 in glioblastoma CSCs, pterostilbene, a plant-derived stilbenoid and dietary component, impaired self-renewal and radioresistance in glioblastoma CSCs through modulating GRP78 signaling and the GRP78/miR-205 axis [234]. Another plant-derived compound, isoliquiritigenin (a chalcone-type bioflavonoid), was shown to suppress both the self-renewal ability and the expression of cancer stemness markers, such as CD44 and ALDH1, in CSCs from oral squamous cell carcinoma; in addition, the invasive and metastatic capacities became impaired in isoliquiritigenin-treated oral CSCs [233]. The anti-stemness effects of isoliquiritigenin were associated with the downregulation of GRP78 (both intracellular and membrane-bound GRP78), whereas those effects could be reversed by GRP78 overexpression [233]. In turn, ovatodiolide (a macrocyclic diterpenoid compound) was able to impair the GRP78 upregulation associated with YAP1 expression, which confers the CSC phenotype in hepatocellular carcinoma cells [228]. Overall, such anti-GRP78 activity of the natural compounds [228,233,234] seems pretty beneficial because they may be used as low-toxicity bioavailable agents for in vivo targeting of CSCs. In this connection, ovatodiolide is noteworthy, as besides its anti-GRP78 activity [228], this natural substance may also decrease HSP27 expression in CSCs [156].

Some small molecule inhibitors of GRP78 were also tested in CSC-related models [80,238,239]. In particular, the known GRP78 inhibitor HA15 (N-[4-[3-[[[5-(Dimethylamino)-1-naphthalenyl]sulfonyl]amino]phenyl]-2-thiazolyl]-acetamide), if combined with PRDM14 knockdown, diminished the fraction of CSC-like SP cells in breast cancer HCC1937 cells [80]. Zebularine, a cytidine analog and inhibitor of DNA methylation, was shown to suppress stemness manifestations in colonosphere-forming colorectal cancer HCT116 cells by downregulating GRP78, while upregulating CHOP, a pro-apoptotic factor [238]. Another inhibitor of GRP78, ruthenium(II) triazine complex [Ru(bdpta)(tpy)]^2+^, reduced the GRP78 protein level in CD133+ CSCs derived from human colon cancer HCT-116 cells; the inhibitor-induced GRP78 downregulation was correlated in a dose-dependent manner with the in vivo-achieved antitumor effects toward tumor xenografts [239]. The cited findings [80,238,239] confirm the rationale of further development of small molecule inhibitors of GRP78 to target cancer stemness.

Intriguingly, there are approaches to targeting intracellular GRP78 by means of the delivery of GRP78 siRNA into target cells. Shen et al. used high-capacity nanoporous silicon carriers to introduce GRP78 siRNA into human breast cancer MDA-MB-231 cells and achieved an 83% reduction of GRP78 gene expression [240]. In a more recent study, 1,2-dioleoyloxy-3-trimethylammoniumpropane (DOTAP) vector-directed liposomes were constructed and used for co-delivery of camptothecin and GRP78 siRNA into CSCs [241]. Compared to the cytotoxic effects of free camptothecin, DOTAP-camptothecin-GRP78 siRNA delivery significantly enhanced CSCs’ sensitivity to the drug [241]. The latter approach seems very promising, as it enables the GRP78-conferred drug resistance of CSCs to be overcome and increases the efficacy of antitumor chemotherapy by means of ‘gene therapy’.

Taking into consideration the pivotal role of GRP78 as the ER stress sensor and main regulator of the UPR, in which there is a balance between cell survival and cell death [43,211], one can suggest this chaperone one of the most vulnerable determinants of cancer stemness.

#### 4.2.2. Extracellular (Plasma Membrane-Bound or Secreted) GRP78

Back in 2010, head and neck cancer cells carrying GRP78 at their surface were characterized as CSC-like cells exhibiting self-renewal and radioresistance [213]. Later, in a murine model of ovarian cancer, fractions of ascites cancer cells expressing GRP78 on the plasma membrane were also identified as self-renewing CSCs with an enhanced spheroid-forming capacity and higher tumorigenicity as compared with ascites cancer cells whose plasma membrane did not contain GRP78 [242]. Zhang et al. showed that endogenous GRP78 expressed on the cell surface was shown to facilitate the adhesion and invasion of hepatocellular carcinoma cells along with enhancement of the secretion and activity of MMP2 [243]. In the same study, cell surface GRP78 expression was inversely correlated with E-cadherin expression, while positively correlating with N-cadherin expression; these observations allowed the authors to suggest the implication of cell surface-bound GRP78 in EMT regulation [243]. It was revealed in a more recent study that GRP78 interacts with CD44v and is co-localized with it on the surface of tamoxifen breast cancer cells; these interactions regulate both CD44v integration into the plasma membrane and CD44v-mediated cell spreading [244]. If an analogous mechanism acts in vivo in GRP78- and CD44v-enriched CSCs, this may facilitate their invasion and metastatic dissemination.

Besides, plasma membrane-bound GRP78 has been found in CSCs from oral squamous cell carcinoma [233] and in CSCs from head and neck cancers [245]. It is asserted in the latter reference that cell surface-associated endogenous GRP78 is a determinant of cancer stemness development, which can be reversed by the GRP78 interactome protein, progranulin: Being expressed on the cancer cell surface, GRP78 is conducive to cancer stemness, but interactions of GRP78 with progranulin, if they occur, retain a non-stem cancer cell phenotype [245]. These findings reveal the unique role of cancer cell surface-associated GRP78 as the ‘deterministic’ driver of the reversible phenotypic modulations taking place in cancer cells. At present, cell surface-associated GRP78 is thought to be one of the established cell surface markers of CSC-like cells [246,247]. In a recent publication [247], Conner et al. described that the GRP78 expression on the surface of breast cancer cells is clearly associated with their reprogramming toward the stem phenotype with high metastatic potential: The tumor-initiating CD24−/CD44+ cells, whose surface was “marked” by GRP78, exhibited an increased propensity to metastasis formation in vivo.

Interestingly, tumor cell-secreted GRP78 was shown to induce the differentiation of bone marrow mesenchymal SCs into cancer-associated fibroblasts (CAFs); such CAF-generating differentiation seemed to be due to activation of the TGFβ/Smad signaling pathway by external (tumor-derived) GRP78 [248]. As CAFs produce growth factors and components of the extracellular matrix contributing to tumor progression, EMT induction, and CSC niche formation [2,3,4,5], the GRP78-conferred increase in the CAF pool within the tumor microenvironment may be favorable for cancer stemness development. It was also found that extracellular GRP78, being secreted by malignant cells, can be entrapped by nearby macrophages through Ajuba receptor-mediated endocytosis pathways [249]. This internalized GRP78 is able to be incorporated into the ER and mitochondrion of macrophages [249]; such an important regulatory and sensor protein as GRP78, as it enters the organelles, may modulate the phenotype and behavior of macrophages within the tumor microenvironment. Taking into consideration that M2 macrophage polarization and some macrophage-derived cytokines (e.g., IL6, IL8, TNFα) stimulate EMT in tumors and CSC niche formation [3,5,9], one can suggest that tumor cell-secreted GRP78 promotes cancer stemness via macrophage-involving mechanisms. Thus, both references [248,249] suggest potential ways by which tumor cell-secreted GRP78 helps to recruit stromal cells (mesenchymal SCs, CAFs, macrophages) for cancer stemness development and tumor progression.

In any event, extracellular GRP78 provides wide opportunities for a multitargeted attack on cancer stemness.

##### Targeting Extracellular GRP78

Anti-GRP78 antibodies were used for targeting cell surface-associated GRP78 in hepatocellular carcinoma cells (Mahlavu and SMMC7721 cell lines); such antibody-conferred neutralization of the endogenous cell surface chaperone led to the inhibition of cell adhesion and invasion [243]. Moreover, targeting cell surface GRP78 with neutralizing antibodies suppressed the secretion and activity of MMP2 and elevated the E-cadherin level, while downregulating N-cadherin [243]. Such results indicate that cancer stemness-associated traits are sensitive to the neutralization of cell surface GRP78, which may be achieved with anti-GRP78 antibodies. 

Using a similar approach, Mo et al. showed that antibodies specifically recognizing the carboxy-terminal domain of GRP78 bind to GRP78 on the plasma membrane of ovarian CSC-like cells and this binding reduces their self-renewal [242]. Both publications [242,243] suggest the possibility of the application of an antibody-based therapy aimed at the neutralization (inactivation) of GRP78 expressed on the surface of CSCs and their precursors.

Isoliquiritigenin (a bioflavonoid), inhibiting the total GRP78 expression in oral cancer CSCs, also reduced the membrane GRP78 level, which was accompanied by downregulation of the expression of LDH1, CD44, and ABCG2, as well as an impairment of invasion and metastatic growth [233]. In this case, isoliquiritigenin may act as a universal agent that simultaneously targets both intracellular and membrane GRP78 in CSCs, thus yielding a dual beneficial effect.

Probably, the development of cell-impermeable inhibitors of GRP78 chaperone activity would also yield effective tools to target CSCs and their precursors (such an approach was discussed toward extracellular HSP90 in Section 3.1.2.).

### 4.3. GRP75

GRP75 (also named mortalin or HSPA9 [41]) largely resides in mitochondria, where it participates in the chaperoning of intramitochondrial proteins and regulation of the organelle work. In addition, the important functions of GRP75 are the control of cell proliferation and also its regulatory interactions with p53, which are connected with the cellular response to genotoxic stresses, radioresistance, apoptotic death, etc. [43,250]. In humans, GRP75 (mortalin-2 or mot-2) is overexpressed in many malignancies and appears to regulate tumor cell propagation and survival [43,250].

The overexpression of GRP75 in hepatocellular carcinoma cells was correlated with vimentin expression and promoted EMT, forming the metastatic tumor cell phenotype [251]. GRP75 overexpressed in breast cancer cells was demonstrated to contribute to EMT by upregulating both the PI3K/Akt and JAK/STAT signaling pathways [252]. In the same study, it was found that GRP75 overexpression was associated with the enhanced expression of mesenchymal markers (vimentin, fibronectin, β-catenin, CK14, hnRNP-K) along with downregulation of epithelial markers (E-cadherin, CK8, CK18) [252]. Similarly, the elevated expression of GRP75 in intrahepatic cholangiocarcinoma cells was associated with EMT induction, which was accompanied by a decrease of the E-cadherin level while increasing the expression of vimentin and Snail; the suppression of GRP75 expression resulted in reverse effects, namely the accumulation of E-cadherin and downregulation of vimentin and Snail expression [253]. The EMT induction associated with GRP75-dependent activation of the Wnt/β-catenin signaling pathway was described for ovarian carcinoma [254] and colorectal cancer [255]. All these references [251,252,253,254,255] provide evidence that GRP75 does promote EMT in cancer cells, thus driving them to a stem (invasive and metastatic) phenotype.

Yun et al. [256] established the implication of GRP75 in cancer stemness by demonstrating on malignant cells of different origins (carcinomas, osteosarcomas, melanoma) that GRP75 overexpression is positively correlated with the upregulation of cancer stemness markers, including ABCG2, OCT4, CD133, CD9, ALDH1, MRP1, and connexin. Besides the upregulated stemness markers, GRP75-overexpressing malignant cells exhibited a greater predisposition to migration and spheroid formation and were less susceptible to a number of antitumor drugs, while GRP75 knockdown with small hairpin RNA (shRNA) sensitized those cells to chemotherapeutics [256]. 

A plant-derived alkaloid, veratridine, was shown to eliminate colon CSCs in a dose-dependent manner, which was correlated with tumor sensitization to 5-fluorouracil and etoposide [257]. The researchers suggested that the CSC-eliminating action of veratridine is due to the drug-induced transactivation of UBXN2A followed by UBXN2A-dependent inhibition of mortalin-2 (GRP75) [257].

Based on such findings, the enhanced expression of tumorous GRP75 may be considered an obligatory factor for the CSC phenotype’s formation/maintenance. Therefore, GRP75 appears to be an extremely attractive target to oppose CSCs and their contribution to cancer pathogenesis. Clinically applicable inhibitors of GRP75 need to be developed for anti-CSC therapy.

### 4.4. TRAP1 (Tumor Necrosis Factor Receptor-Associated Protein 1)

TRAP1 (HSP75 or HSC5 [41])—a 75 kDa member of the HSP90 subfamily—is an ATP-utilizing molecular chaperone localized to mitochondria and participating in the regulation of proliferation, apoptosis, differentiation, and energetic metabolism in both normal and cancer cells and non-stem ones [42,61,258]. 

There are reports about the implication of TRAP1 in cancer cell stemness (see [42,61] for a review). According to the data of Lettini et al. [259], in human colorectal cancer, TRAP1 is co-expressed with stemness markers and contributes to the development/maintenance of the CSC phenotype by activating the Wnt/β-catenin signaling pathway. This cancer stemness-promoting mechanism was based on TRAP1-mediated modulation of the expression of frizzled receptor ligands and β-catenin modification (ubiquitination/phosphorylation), so that TRAP1 upregulated the expression of β-catenin as well as several Wnt/β-catenin target genes [259]. On the contrary, in other investigations performed on human ovarian cancer, TRAP1 downregulation resulted in the enhancement of invasion and EMT [260]; such a discrepancy implies that cancer stemness-related activities of TRAP1 may cardinally differ in various types of tumors.

However, the activities of TRAP1 in glioblastomas seem to be conducive to cancer stemness development [261,262]. It was demonstrated that increased TRAP1 expression is required for proliferation, migration, and neurosphere formation in glioblastoma cells, and also for their resistance to temozolomide chemotherapy, which was associated with TRAP1-mediated metabolic reprogramming [261]. In another publication on a similar topic, cooperative interplay between mitochondrial TRAP1 and the major mitochondrial deacetylase sirtuin-3 in glioma CSCs was reported to be responsible for the reduced production of ROS and plasticity of energetic metabolism in CSCs, thus facilitating their adaptation to hypoxia and nutrient deficiency and maintaining the CSC phenotype [262]. Taking into consideration that (i) cell migration and spheroid formation, (ii) resistance to drugs and ROS, and (iii) adaptiveness to hypoxia and energy starvation are hallmarks of CSCs (see the introduction section), the described activities of TRAP1 in glioblastomas characterize this mitochondrial chaperone as a one of the drivers in cancer stemness development. The TRAP1-mediated reprogramming of the work of mitochondria and energetic metabolism in cancer cells can play an especially important role within hypoxic (poorly vascularized) zones of solid tumors, sites where EMT is induced and CSC generation occurs.

Small molecule inhibitors of TRAP1, such as gamitrinib [263] and gamitrinib-triphenylphosphonium [264], were developed and their antitumor effects are known. However, the responses of human CSCs to the action of those TRAP1 inhibitors remain to be assessed. TRAP1 seems to be a promising target for anti-CSC therapy because inhibiting this chaperone may prevent CSC phenotype formation through EMT, as well as eliminating existing CSCs or sensitizing them to the conventional treatment of cancer. 

### 4.5. Protein Disulfide Isomerases

These redox-dependent chaperones are members of a superfamily of ubiquitous oxidoreductase proteins largely localized to the ER but also residing in the cytosol, nucleus, mitochondria, and on the cell surface or being secreted, whose main function is to maintain proteostasis. Protein disulfide isomerases catalyze the formation or breaking/isomerization of intramolecular S-S links in cysteine-containing proteins, thus performing mandatory reactions in the protein folding machinery [47]. Protein disulfide isomerases are known to be implicated in oncogenesis and their expression was found to be elevated in various types of malignancies, which correlated with poor patient outcomes and enhanced tumor invasion, metastases, and failure of chemotherapy [47].

There are data indicating an involvement of protein disulfide isomerases in the development and maintenance of cancer stemness. A proteomics-based study identified protein disulfide isomerase (along with GRP94 and GRP78) among proteins whose overexpression is associated with the metastatic spread of cancer cells [206]. Later, it was found that such a member of the protein disulfide isomerase family as P4HB is overexpressed in hepatocellular carcinoma cells and promotes EMT in them [227]. The elevated level of another protein disulfide isomerase, anterior gradient 2 (AGR2), was shown to drive EMT in prostate cancer, which was accompanied by the formation of the aggressive tumor cell phenotype with increased invasive, pro-metastatic, and angiogenic potentials [265]. The EMT-stimulating effect of AGR2 in prostate cancer cells was explained by AGR2-conferred stabilization of protein p65, which, as a result, was able to activate NFκB, thereby promoting EMT [265]. In addition, AGR2 appears to be one of the key players in forming the tumor niche and remodeling the tumor microenvironment to promote malignant growth with metastasis spread [266].

Importantly, protein disulfide isomerases can also enable some manifestations of cancer stemness that are critical for the pathogenesis. For instance, overexpression of an aberrantly spliced isoform of AGR2, AGR2vH, in cholangiocarcinoma cells was correlated with vimentin expression and enhanced cell migration, invasion, adhesion, as well as moderate proliferative activity, which are all signs of a CSC-like (pro-metastatic) phenotype; AGR2vH knockdown with an isoform-specific RNAi attenuated these CSC phenotype-associated features [267]. In a breast cancer model, the AGR2 levels were found to become increased in response to hypoxia, which was due to Twist1 binding to an E-box sequence within the AGR2 promoter and triggering of AGR2 gene transcription [268]. The discovered mechanism of Twist1-mediated upregulation of AGR2 seems especially important as it is AGR2 that is required for Twist1-induced activation of breast tumor cell migration and invasion, which are associated with cancer stemness [268]; thus, the AGR2-Twist1 axis should be considered as one of the endogenous drivers in cancer stemness development. There were findings indicating the implication of ARG2 in maintaining the stem phenotype in colorectal CSCs in which this chaperone has been identified as a stem cell marker [269]. In this study, Wnt/β-catenin pathway-regulated AGR2 expression was shown to be correlated with the expression of known cell surface stem markers as well as with the cell spheroid-forming capacity [269]. Other researchers have demonstrated that AGR3 is also implicated in stemness development/maintenance in colorectal cancer via AGR3-mediated modulating of the Wnt/β-catenin signaling pathway; notably, these stemness-promoting and Wnt/β-catenin pathway-modulating activities of AGR3 were dependent on the presence of frizzled 4 (FZD4), a G-protein-coupled receptor for Wnt proteins [270]. It seems rather intriguing that the development and maintenance of cancer stemness may be based on the FZD4-dependent regulatory axis involving two protein disulfide isomerases (AGR3-Wnt/β-catenin signaling-AGR2), as it follows from the studies performed on colorectal cancer [269,270]; at least, this may provide additional opportunities for he therapeutic targeting of cancer stemness.

Extracellular (secreted) protein disulfide isomerases may also promote the stemness-associated activities of cancer cells. In the case of prostate cancer, secreted AGR2 was shown to stimulate angiogenesis by enhancing VEGF receptor 2 activity and facilitate metastasis spread [265]. In turn, extracellular AGR3 participates in the regulation of the adhesion and migration of breast cancer cells via activation of Src kinase-driven signaling [271]. Taking into account that high cell motility and migration are some of the characteristic features of CSCs and associated with tumor invasion and metastases, pharmacological inhibition of the secretion/activities of extracellular protein disulfide isomerases appears to be one more option for targeting chaperone-conferred cancer stemness and its contribution to the pathogenesis.

Clinically applicable inhibitors of protein disulfide isomerases are obviously needed to target CSCs in patients’ tumors. The development of various small molecule inhibitors of protein disulfide isomerases is being continued [272] and the newly created variants should be tested in cancer stemness-relevant models. 

### 4.6. Calreticulin

This Ca^2+^-binding and Ca^2+^ level-regulating chaperone is preferentially localized to the ER but sometimes to the cell surface as well [44,45]. It is generally accepted that calreticulin is one of the cancer-promoting proteins because its high expression was shown to correlate with active cell proliferation in different types of cancer, with metastasis occurrence and with VEGF-stimulated angiogenesis [44,45]. Meanwhile, being exposed on the cell surface or secreted, calreticulin can form complexes with ERp57 and integrins, and these interactions regulate signals for immunogenic death or phagocytosis of malignant cells [273]. Therefore, calreticulin is regarded as a potential target/tool for antitumor therapy. 

According to one publication of 2009, upregulated calreticulin was found under suppression or reversal of EMT in breast cancer cells re-expressing kallikrein-related peptidase 6 [274]. However, three works were published in which overexpression of calreticulin promoted EMT in different tumor cells [275,276,277]. So, in a model of gastric cancer, calreticulin knockdown with siRNA suppressed TGF-β1-induced EMT in the tumor cells as well as impairing their migration, invasion, and metastasis formation in vivo [275]. In another study, it was shown that calreticulin overexpressed in lung cancer A549 cells significantly activated TGF-β1-induced EMT in a calcium-dependent manner; this mechanism was realized via calreticulin-mediated modulation of both Smad signaling and Ca^2+^ influx and Ca^2+^ signaling [276]. The other mechanism of calreticulin-mediated regulation of EMT was revealed in pancreatic cancer cells: Calreticulin can promote EGF-induced EMT by enhancing cell migration/invasion and decreasing E-cadherin; it was due to calreticulin-conferred activation of the integrin/EGFR-ERK/MAPK signaling pathway, which includes EGFR phosphorylation at tyrosine 1173 [277]. In the same study, calreticulin was found to be co-localized with pEGFR1173, fibronectin, and integrin β1, while being co-immunoprecipitated with fibronectin, integrin β1, and c-Myc, which suggests the direct involvement of calreticulin in the regulation of EGF-induced EMT in pancreatic tumors [277]. In support of this, there are findings indicating significant accumulation of calreticulin in pancreatic CSC-like cells [278]. It was shown that the high level of cell surface calreticulin can be used as a biomarker of pancreatic CSC-like cells, particularly their SP cell fraction (i.e., dye-excluding cells with actively working ABC transporters) [278].

Notably, one of the mutations driving the myeloproliferative neoplasm (MPN) development is that of the calreticulin-encoding CALR gene; this mutation activates MPL/JAK/STAT signaling in MPN stem cells, thereby promoting better survival and expansion outside the bone marrow niche [279]. Therefore, in the case of MPN, mutated calreticulin may be responsible for the development and maintenance of malignancy-associated stemness. The recently discovered function of calreticulin in cancer cells is preventing the p53-mediated caspase-independent cell-death response; the latter can be induced by calreticulin knockdown, leading to mitochondrial Ca^2+^ overload followed by cell killing through a mitochondrial permeability transition pore-dependent mechanism [280]. It seems likely that an analogous function of calreticulin is also performed in calreticulin-enriched CSCs to protect them from p53-mediated cell-death response induction and thus increase their stress resistance/adaptiveness.

Apparently, calreticulin may be used as a target or a tool to oppose cancer stemness. On the one hand, inhibition of the functional activities of calreticulin may suppress EMT and attenuate the most dangerous properties of CSCs, such as their high motility, capacities to migration, invasion and metastatic spread, drug resistance etc. On the other hand, it seems possible to use calreticulin for the eradication of CSCs via immunogenic cell death. In vivo, the augmented expression of calreticulin on the surface of cancer cells usually triggers a calreticulin-dependent mechanism of their immunogenic killing by cytotoxic T-lymphocytes [273]. Therefore, an important task is to force CSCs to express more calreticulin on their plasma membrane in order to provoke T-lymphocytes’ attack against them. 

The expression/behavior of calreticulin in CSCs and cancer cells undergoing EMT (i.e., CSC precursors) may be modulated by means of small molecule compounds. For example, honokiol, a plant-derived biphenol, was shown to induce both calpain-mediated degradation of calreticulin and early implication of calreticulin in immunogenic cell death; actually, the compound inhibited EMT and metastasis formation while stimulating the immunogenic killing of tumor cells in a murine model of gastrointestinal cancer [275]. 

In some cases, calreticulin appears to help overcome the high radioresistance intrinsic to CSCs. Gameiro et al. found that CSCs, being resistant to the direct cytolytic action of proton beam irradiation, exhibited a post-irradiation increase in calreticulin expressed on the cell surface, which rendered them vulnerable to CTL-mediated killing [281]. Taken together, these data [275,281] confirm the idea that calreticulin may be both a target and a tool for anti-CSC therapy.

## 5. Conclusion and Perspectives

Summarizing all the reviewed data, one can conclude that molecular chaperones are really the determinants of cancer stemness, which, on the one hand, define the formation of the CSC phenotype and, on the other hand, maintain and mediate the most critical properties of CSCs, thus making them the drivers of cancer pathogenesis. Literally, any manifestations of cancer stemness and almost all features of CSCs are causally connected with the functioning of molecular chaperones. HSPs, GRPs, peptidyl-prolyl isomerases, protein disulfide isomerases, and calreticulin are implicated in EMT induction and the design of the CSC phenotype via the modulation of gene transcription and signaling pathways, reprogramming of energy metabolism and work of organelles, altering composition and responses of the cytoskeleton and cell membrane/surface, enhancing the expression and secretion of extracellular proteinases, etc. As a result of such activities of molecular chaperones, CSCs become what they represent, namely tumorigenic, mobile, and invasive fibroblast-like cells, which express a set of specific markers and can renew themselves, are able to migrate and form spheroids and distant metastases, are resistant to stressful factors of the microenvironment and are able to remodel their microenvironment into cancer-promoting niche, and are resistant to chemotherapy and radiotherapy, thus being a cause of tumor recurrence and poor patient outcomes. 

Here, it should be added that an extracellular sojourn of some chaperones, such as HSP90, HSP70, GRP78, AGR2, AGR3, and calreticulin, also contributes to the cancer stemness development and manifestation of characteristic features of CSCs. These extracellular (cell surface-associated or secreted) chaperones can regulate secretion processes, MMP activities, and migration of CSCs as well as their invasive, metastatic, and angiogenic potential. Besides, CSC-secreted chaperones appear to facilitate remodeling of the tumor microenvironment into the CSC niche.

Taken together, these data allow us to consider cancer stemness development as a chaperone-dependent process and CSCs as tumorigenic cells being extremely addicted to molecular chaperones. The latter are in fact necessary for both the emergence of CSCs through EMT or similar phenotypic modulations and the manifestation of cancer stemness-associated hallmarks, including the high motility and invasiveness of CSCs, their tendency to metastatic spread, their elevated resistance to chemotherapy and radiotherapy, etc. Consequently, artificial inhibition of chaperone activity or the expression of chaperones may prevent the generation of new CSCs and make existing CSCs more sensitive to therapeutics.

Interestingly, among all the chaperones considered in this review, there were only two endogenous chaperones, namely FKBPL and FKBP12 (both are members of the immunophilin family), whose activities interfered with the development of CSC phenotype [178,179,181]. FKBL-derived peptides, AD-01 and ALM201, were shown to suppress cancer stemness in vitro and in vivo [178,179,180]; ALM201 is currently undergoing clinical trials as an antitumor agent [180]. 

The many dozens of publications cited in this review demonstrate in various models in vitro and in vivo that inhibitors of the activity or expression of chaperones can (i) prevent (or even reverse) EMT and similar phenotypic modulations toward cancer stemness; (ii) suppress the most critical manifestations of cancer stemness, such as self-renewal, migration, invasion, and metastatic behavior of CSCs; and (iii) sensitize CSCs to chemotherapy and radiotherapy. Many chaperone-targeting agents have been tested in preclinical and clinical trials for more than a decade. Nevertheless, no therapeutic modalities based on the specific inhibition of chaperones have been developed so far. Potent and selective inhibitors of chaperones are still missing from the pharmaceutical arsenal of oncologists. The authors hope that the reading of the present review will motivate molecular pharmacologists to further their efforts on the development of clinically applicable methods of targeting chaperones in CSCs and CSC precursors (i.e., cancer cells undergoing EMT and thus acquiring a stem-like phenotype). In this connection, the most promising approaches are as follows:

(1) The creation and use of highly selective inhibitors that disrupt the interaction of a chaperone with only one client protein or co-chaperone, or co-factor, provided that this interaction is critical for the development and maintenance of the CSC phenotype; such a selectivity, if is achieved, will help to alleviate the toxic effects on normal tissues and organs. Perhaps the use of computer-aided drug design will enable better advancement of this direction [59]. (2) Creation and use of hypoxia- and acidosis-sensitive inhibitors of chaperones, i.e., chaperone-targeting drugs whose inhibitory properties become activated only in the context of low oxygen and/or low pH; exactly upon such conditions, many CSCs are generated as a result of EMT occurring within the poorly vascularized (hypoxic) regions of solid tumors. This approach will help to elevate the selectivity of the cytotoxic effects of chaperone-targeting drugs toward CSCs and their precursors. (3) Prevention of the induction of new cytoprotective chaperones in CSCs treated with inhibitors of the chaperone activity or hyperthermia, or proteasome inhibitors. This will enable an escape of the problems outlined in Figure 5 and Figure 6, when after the inhibitory treatment or hyperthermia, the target cells may survive and become more resistant and adapted to therapeutics. (4) The use of inhibitors of chaperones residing within the ER and mitochondria to modulate the UPR and functioning of the mitochondrial permeability transition pore in CSCs in a manner of causing their death or sensitization to therapeutics. (5) The use of cell-impermeable inhibitors of extracellular chaperones: Such chaperone-targeting agents can be much less toxic and their beneficial effects cannot be impaired through the high activity of drug-excluding membrane ABC transporters, which is typical for CSCs. Probably, the cell-impermeable inhibitors of chaperones will not kill the target cells (CSCs or their precursors), but such inhibitors are quite able to attenuate the stemness-related activities of CSCs (invasion, metastasis spread, etc.) and also sensitize them to drug or radiation exposure. (6) The use of liposomes, nanoparticles, or artificially prepared exosomes as vector-directed vehicles for targeted delivery of chaperone-suppressing agents into CSCs and their precursors: Thanks to such an approach, it will be possible to introduce into cells certain tools of ‘gene therapy’, e.g., siRNAs, or microRNAs, or antisense-based constructs for knockdown of either chaperone. (7) The creation and use of inhibitors of chaperones that would enable prevention of the immune selection of CSC-like phenotypes and sensitization of CSCs to immune attack and regimens of immunotherapy. (8) A further search for suitable natural inhibitors of chaperones in human CSCs (it is the modern trend to use natural compounds in the fight against cancer). Such a focused search should cover a wide spectrum of biological products from plant-derived small molecule substances to microRNAs of various origins. 

Finally, there is an intriguing opportunity to transform the cancer stemness-promoting chaperones from a serious challenge for antitumor therapy into a potent remedy of antitumor therapy. For example, the drug- or radiation-induced translocation of calreticulin to the surface of CSCs may result in their immunogenic death and, respectively, therapeutic benefits [275,281]. A non-standard and promising approach was described by Chinese researchers, who propose not to inhibit chaperones in CSCs but, instead, to isolate from CSCs chaperone-peptide complexes for preparation of effectual vaccines against cancer [282]. 

## Figures and Tables

**Figure 1 cells-09-00892-f001:**
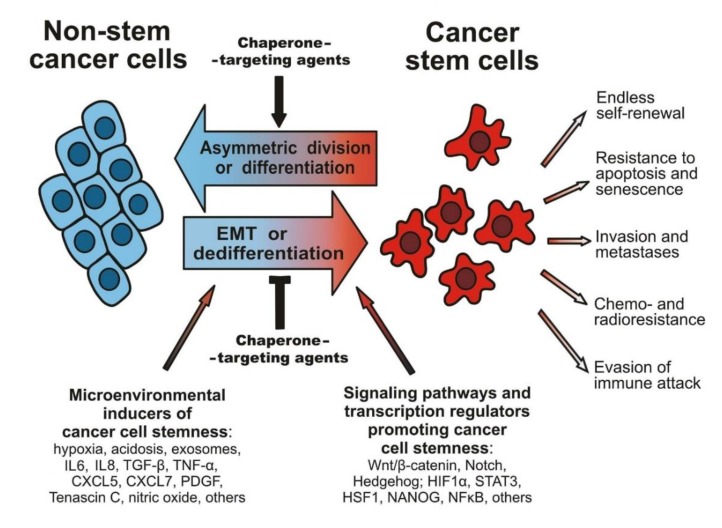
A scheme illustrating the plasticity of cancer cell phenotypes and reversible transitions between the pools of non-stem cancer cells and CSCs. The figure includes extracellular and intracellular stimuli, factors, and pathways that can contribute to the cancer stemness development, and also shows the characteristic CSC properties that are critical for the tumorigenicity and cancer pathogenesis. Potential opportunities for chaperone-targeting agents are here indicated by black arrows (see details in the text).

**Figure 2 cells-09-00892-f002:**
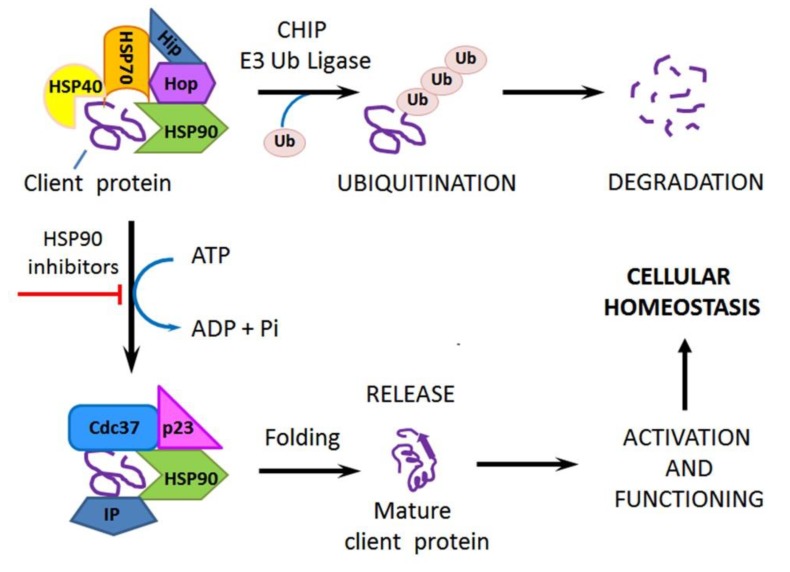
A simplified scheme showing the work of HSP90 chaperone machine. This ATP-dependent chaperoning is carried out by the two major chaperones HSP90 and HSP70 in cooperation with several co-chaperones and co-factors, including HSP40, Hip, Hop, cell division cycle control protein 37 (Cdc37), p23, and immunophilins (IP) (adapted with some modifications from Kabakov et al. [49]). For different client proteins, a set of the assisting co-factors may somewhat vary and include other components [50]. It is seen from the scheme that a client protein may end up in two alternative outcomes of the chaperoning: folding and maturation or ubiquitination and degradation. The latter path can be promoted by inhibitors of HSP90 activity [49,50,51].

**Figure 3 cells-09-00892-f003:**
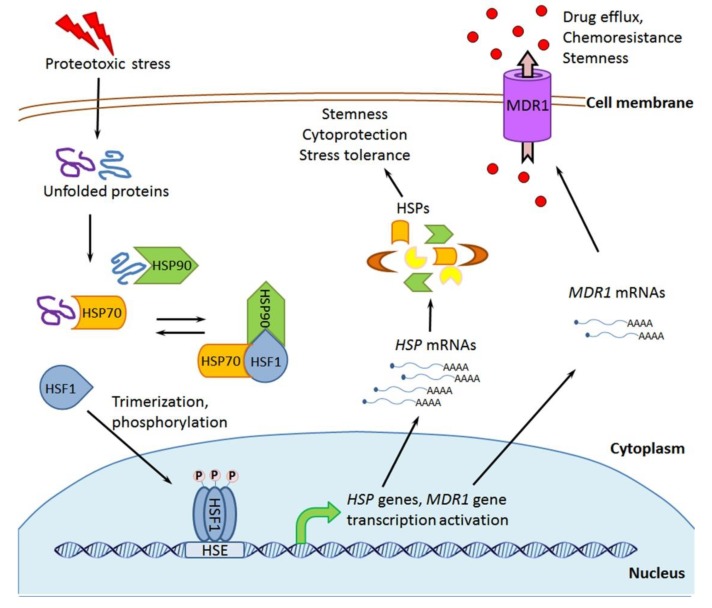
A scheme showing a mechanism of the HSF1-mediated heat stress transcription response resulting in the expression of heat shock protein (HSP) and multidrug resistance 1 (MDR1) genes whose protein products contribute to cytoprotection from stresses, drug resistance, and also to the cancer stemness. Cytosolic HSP90 and HSP70 bind to and passivate HSF1, thus preventing its activation under normal conditions. When stress-damaged proteins accumulate in the cytoplasm, they recruit HSP70 and HSP90 from the inactivating complex with HSF1, so that the latter is activated via its phosphorylation and trimerization, then binds to the heat shock element (HSE) sequence in promoter regions of HSP genes, and thus triggers the expression of all inducible HSPs. As the MDR1 gene is known to have the HSE-containing promoter, MDR1 expression can also be initiated by activated HSF1 [29].

**Figure 4 cells-09-00892-f004:**
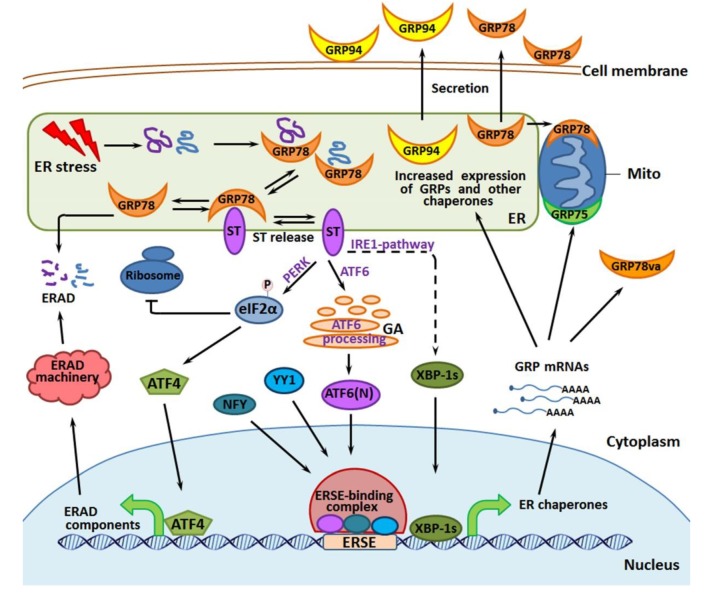
A simplified scheme showing the endoplasmic reticulum (ER) stress response: GRP78 residing within the ER lumen is bound to ER-specific stress transducers (STs) to maintain them in the inactive state. In the case of ER stress, GRP78 is recruited by stress-damaged (misfolded) proteins, which is accompanied by the release of three STs, namely protein kinase RNA-like endoplasmic reticulum kinase (PERK), activating transcription factor 6 (ATF6), and inositol-requiring enzyme 1 (IRE1). These STs are activated and trigger the so-called ‘unfolded protein response’ (UPR), which stimulates certain signaling pathways, resulting in the arrest of translation while generating the active form of ATP4. In the nucleus, active ATP4 initiates the expression of genes whose products are components of the ER-associated protein degradation (ERAD) machinery aimed at the disintegration of misfolded proteins. After its processing in the Goldgi apparatus (GA), ATF6 and the spliced form of X-box binding protein 1 (XBP-1s), acting together with transcriptional factors Yin Yang 1 (YY1), nuclear factor Y (NF-Y), and chromatin modifiers, form the ER stress response element (ERSE)-biding complex that activates the ERSE-containing promoters of ER stress-responsive genes. GRPs are major products of the UPR, which primarily serve for the refolding of stress-damaged proteins within the ER and mitochondria (Mito). Additionally, GRP94 and GRP78 can be relocalized to the cell membrane or secreted, while a cytosolic isoform of GRP78 (GRP78va) can be generated as a result of alternative splicing (see [43] for details).

**Figure 5 cells-09-00892-f005:**
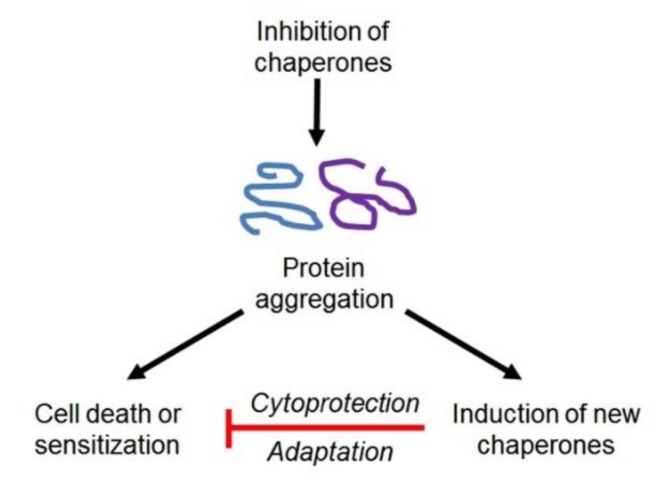
A simplified scheme outlining an undesirable situation that may take place under the use of cell-penetrating inhibitors of the chaperone activity against tumors. The dysfunction of intracellular chaperone(s) will result in the accumulation and aggregation of misfolded proteins, which will trigger the induction of new cytoprotective chaperones, thus adapting target tumor cells and rendering them more resistant to cytotoxic drugs and radiation exposure. This problem may especially be actual for functional inhibitors of cytosolic HSP90 or HSP70, whose inactivation can provoke the HSF1-mediated expression of inducible HSPs and also MDR1 (see the text for details).

**Figure 6 cells-09-00892-f006:**
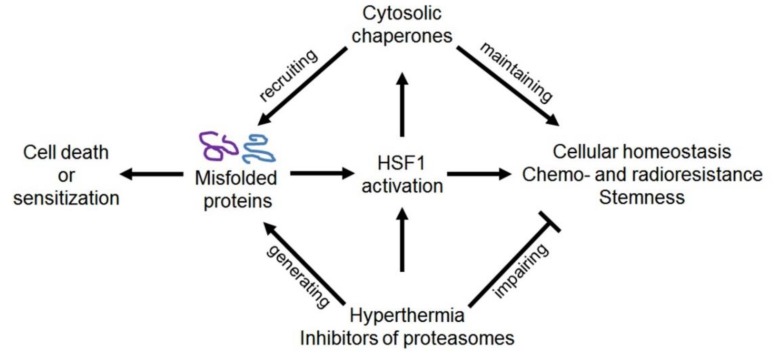
A scheme showing the duality of the action of hyperthermia or proteasome inhibitors on CSCs. The treatment-induced intracellular accumulation of misfolded proteins depletes the cytosolic pool of chaperones, so that the chaperone-dependent maintenance of the cell viability and cancer stemness is ceased, which may yield the elimination or sensitization of treated CSCs. However, the excess misfolded proteins activate HSF1, which triggers HSP expression and then the new chaperones (HSPs) will promote the survival and recovery of the treated CSCs.

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
