# Peer review of "Molecular Chaperones in Cancer Stem Cells: Determinants of Stemness and Potential Targets for Antitumor Therapy"

_cells, 2020, doi:10.3390/cells9040892_

Round 1
Reviewer 1 Report
In this article, the authors address the potential role of molecular chaperones in cancer stem cells. Some aspects of cancer cell stemness are explained, including those related to transcriptional regulation and some signalling pathways. Even though various topics are well addressed, others are only partially addressed, such that some key information is missed. For example, the development of novel small molecules against HSPs. Other topics are not addressed at all, for example, the use of immunophilin ligands in cancer treatment, the role of HSP90-FKBP complexes in the cell trafficking of key factors such as GR or NF-kB. Others are totally ignored, even when the main title is indeed mentioned, for example telomerase: the role of Hsp90 and FKBPs in telomerase activity is not even mentioned. Similarly, the various and often opposite effects mediated by FKBP51 according to the cell type (normal fibroblast versus cancer cell) or the antiapoptotic action of the immunophilin have not been considered. The role of HSPs and immunophilins in cell differentiation has not been addressed either. In short, the overall article is good and potentially interesting for a broad spectrum of readers, its main defect being the fact that sometimes it overwhelms the reader with a lot of information that is not entirely connected with the main topic of the work, whereas related topics that may fit better in its aims are not addressed at all. I would suggest a deep revision of the manuscript. I feel that it is potentially worthy, but it should be polished according to the above-listed observations.
Author Response
Response to Reviewer 1:
Dear Sir / Madam,
Thank you very much for reviewing our manuscript and helpful criticism.
Below we are trying to respond to your remarks point-by-point. In the revised version, we marked by yellow all new text fragments which were inserted to attenuate your criticism.
- You wrote: “Even though various topics are well addressed, others are only partially addressed, such that some key information is missed. For example, the development of novel small molecules against HSPs.” In the revised version, we have added a new reference [59] that is a rather fresh review dedicated to the development of novel small molecule inhibitors of HSPs (please see pages 9, 38 and 41 – marked by yellow). We are not quite agreed that this point (development of novel small molecule inhibitors of HSPs) was missed in our manuscript. There are many published reviews with description of inhibitors of HSPs and their potential use against cancer. However, the originality of our review is based on the summarizing analysis of inhibitors of HSPs (chaperones) which were shown to target CSCs or mechanisms of CSC generation (EMT), or manifestations of some (cancer-aggravating) properties of CSCs. That is why we tried to be focused on HSP-inhibiting drugs and compounds (most of which are the small molecule ones) that targeted CSCs or suppressed the CSC phenotype-promoting pathways. We did not include inhibitors of HSPs if there were not publications about their effects of CSCs or EMT (effects of such inhibitors on non-stem cancer cells were not the subject of our review). Nevertheless, we here give several fresh refs where the problem of development of new inhibitors of HSPs is reviewed. Besides the newly added ref [59], these are: [51], [52], [58], [62], [114], [137] and several others (for non-HSP chaperones). It seems to us that our review covers a huge volume of the CSC-relevant information on inhibitors of HSPs (including the small molecule ones).
- You wrote: “Other topics are not addressed at all, for example, the use of immunophilin ligands in cancer treatment,…” Thanks for this remark. In the revised version, we have included additional (CSC-relevant) information on the role/effects of FKBPL and FKBPL-derived peptides with mentioning of current clinical trials of one of the peptides as a potential anticancer agent: see the yellow-marked inserts on pages 25, 27, 28, 37 and 48, including the three newly added (“yellow”) refs [178-180]. We think that the addition of this new and relevant material has improved our manuscript and you will appreciate it.
- You wrote: “…the role of HSP90-FKBP complexes in the cell trafficking of key factors such as GR or NF-kB…” Thanks for your remark. Yes, this interaction (HSP90-FKBP) is important for trafficking of steroid receptors and NF-kB that may be key events in some aspects of cell regulation. However, we have not found in PubMed/MEDLINE any publications directly establishing the importance of the HSP90-FKBP complexes for the CSC phenotype or EMT. That is why we only discuss such a possibility in the revised version: please see the yellow-marked text fragments on pages 14 and 27.
- You wrote: “…the role of Hsp90 and FKBPs in telomerase activity is not even mentioned.” Thanks for your remark but we think that you are not quite correct. The previous version of our manuscript contained mentioning of the role of HSP90 in the telomerase (hTERT) activity (which is one of client proteins of HSP90); you could see it in the beginning of subsection 3.1.1. of the old version. In the revised version, please see the yellow-marked text fragments on pages 11, 14 and 27; there we discuss a potential significance of the FKBP-HSP90-telomerase (hTERT) interactions for the CSC phenotype. As we have not found publications establishing such a significance for CSCs or EMT, we do not describe the FKBP-HSP90-telomerase (hTERT) interactions in more detail, trying to escape the excessive speculations.
- You wrote: “Similarly, the various and often opposite effects mediated by FKBP51 according to the cell type (normal fibroblast versus cancer cell) or the antiapoptotic action of the immunophilin have not been considered.” Thanks for your remark. The diversity of FKBP51-mediated effects in normal fibroblasts and cancer cells is an interesting issue indeed, but this is beyond the topic of our review article. Here we cite and analyze only those (FKBP51-mediated) effects which are important for CSCs or EMT (see the subsection 3.6.). The role of immunophilins in regulation of apoptosis is now mentioned in the revised version: see the yellow-marked fragments on pages 25 and 27.
- You wrote: “The role of HSPs and immunophilins in cell differentiation has not been addressed either.” Thanks for your remark. Yes, HSPs and immunophilins play an important role in cell differentiation but this is beyond the subject of our review. (Here we consider EMT and similar CSC-generating phenotypic rearrangements which are rather dedifferentiation than differentiation, and contributions of HSPs and other chaperones including immunophilins to these processes are reviewed in detail.) In the revised version, the important role of HSPs and/or immunophilins in cell differentiation is mentioned on pages 11 and 25 (marked by yellow).
Thanks for your consideration.
With best wishes,
On behalf of all the authors,
Alexander Kabakov
Reviewer 2 Report
The subject of is very intriguing and current topic. The authors clearly summarize the principal papers concerning the subject of the review.
Author Response
Response to Reviewer 2:
Dear Sir / Madam,
Thank you very much for reviewing our manuscript and favorable estimation.
With best wishes,
On behalf of all the authors,
Alexander Kabakov
Reviewer 3 Report
I reviewed the manuscript by Kabakov et al. entitled: „Molecular Chaperones in Cancer Stem Cells: Determinants of Stemness and Potential Targets for Antitumor Therapy”.
This immense manuscript summarizes data reffering to the role of molecular chaperones in cancer stem cells. The topic is rather interesting. So far there are a lot of reports devoted for study moleculare chaperone, among others Heat Shock Proteins (HSPs), in cancer. The strentgh of this manuscript is overviewing introduction describing the main characteristics of CSCs and Molecular Chaperones. The other one relates to description of effects of extracellular HSPs on CSC phenotype.
However, I have some comments:
- I propose to divide the first Chapter „Overviewing introduction” into two parts (1) – general description of CSCs and (2) A contribution of EMT to CSCs phenotype.I think it may make understanding easier in later chapters describing the role of HSPs in CSCs.
- The second comment corresponds to nomenclature of HSPs. I suggest to use guidelines for the nomenclature of the human HSP families, HSPH (HSP110), HSPC (HSP90), HSPA (HSP70), DNAJ (HSP40), and HSPB (small HSP), which were described in the manuscript of Kampinga H et al. (2009). As it was mentioned by Kampinga et al. (2009) there is huge number of members in the various human HSP families. Hence, inconsistencies in their nomenclature have led to confusion and misunderstanding of work results. For example, almost identical names were used for HSP70-2, which is a protein product of HSPA1B gene, whereas HSP70.2 correspond s to the testis specific HSPA2 membre. The authors also use name of HSP70-2 for HSPA1B gene in text and can provide some misundestanding for readers.
- If it is possible, I propose to summarise the role of molecular chaperones in Cancer Stem Cells in Table as proposed below. I think it additionally clarifies the content of main text.
|
Molecualr Chaperone |
Mechanism of contribution to the CSC phenotypes |
Possible targeting |
References |
|
|
|
|
|
Author Response
Response to Reviewer 3:
Dear Sir / Madam,
Thank you very much for reviewing our manuscript and helpful criticism.
Below we are trying to respond to your remarks point-by-point. In the revised version, we marked by green all new text inserts which we made according to your comments.
- You wrote: “I propose to divide the first Chapter „Overviewing introduction” into two parts (1) – general description of CSCs and (2) A contribution of EMT to CSCs phenotype. I think it may make understanding easier in later chapters describing the role of HSPs in CSCs.” Thanks for your remark.
Yes, we are agreed that the section Introduction should be better structured. In the revised version, we have divided Introduction into 3 subsections (marked by green). This is not the same but similar to what you recommended. We think that this really improves understanding of the next sections.
- You wrote: “The second comment corresponds to nomenclature of HSPs. I suggest to use guidelines for the nomenclature of the human HSP families, HSPH (HSP110), HSPC (HSP90), HSPA (HSP70), DNAJ (HSP40), and HSPB (small HSP), which were described in the manuscript of Kampinga H et al. (2009). As it was mentioned by Kampinga et al. (2009) there is huge number of members in the various human HSP families. Hence, inconsistencies in their nomenclature have led to confusion and misunderstanding of work results. For example, almost identical names were used for HSP70-2, which is a protein product of HSPA1B gene, whereas HSP70.2 correspond s to the testis specific HSPA2 membre. The authors also use name of HSP70-2 for HSPA1B gene in text and can provide some misundestanding for readers.”
Thanks for your remark. We are absolutely agreed and we have made as you recommended. In the revised version, the new reference (Kampinga et al. 2009) has been introduced: [41] – this ref is cited throughout the text of manuscript and respective names of each HSP or GRP are now given according to the proposed nomenclature (marked by green on pages 7, 11-13, 17, 18 and further without color); the requested ref [41] is green within the list of refs (page 40).
- You wrote: “If it is possible, I propose to summarise the role of molecular chaperones in Cancer Stem Cells in Table as proposed below. I think it additionally clarifies the content of main text…”
Thanks for your remark and proposal. At first, we liked your idea and we began to create a draft of the proposed Table. Unfortunately, we have seen that the diversity of mechanisms and pathways, and large amounts of chaperone-targeting agents complicate the task; only HSP90 took about a page, so that we think that the completed Table would be expanded onto 6-7 pages that is too much, of course. So, please allow us not include this Table; it seems easier to find any information in the main text which is well structured (i.e. divided into subsections).
Thanks for your consideration.
With best wishes,
On behalf of all the authors,
Alexander Kabakov
Reviewer 4 Report
The authors provided a very detail review of the involvement of chaperones in the characteristics of cancer stem cells and the therapeutic potential of chaperone-targeting agents. This manuscript is very informative for the researchers working in this field. To make the article focused on chaperones, it is suggested to reduce the descriptions about cancer stem cells in the first section (probably reduce to two pages for the most). In addition, the authors included the surface form of chaperones (such as Hsp70 or Grp78) in this manuscript but without discussing the potential of these surface forms in the development of therapeutic strategies. It is suggested to add the discussion of targeting surface form of chaperones by immunotherapy.
Author Response
Response to Reviewer 4:
Dear Sir / Madam,
Thank you very much for reviewing our manuscript and favorable estimation.
Below we are responding to your comment. In the revised version, we marked by blue the new text fragments which are concern to your remark.
You wrote: To make the article focused on chaperones, it is suggested to reduce the descriptions about cancer stem cells in the first section (probably reduce to two pages for the most). Thanks for your remark. Probably, you are largely right, but it was not our goal – to prepare the article mainly focused on chaperones. We wanted to write a review dedicated to chaperones in CSCs and how it is connected with cancer pathogenesis. For this purpose, the long detailed first section is very important; in fact, nothing superfluous is there: for each feature of CSCs, for each EMT-promoting mechanism or signaling pathway described in ‘Overviewing Introduction’, respective chaperone-dependent contributions are described in the next sections. We are afraid that in the case of shortening of the first section, many key points will become unclear and the concept of our review will be broken. Based on such reasoning, please allow us not to reduce the first section (‘Overviewing Introduction’).
You wrote: In addition, the authors included the surface form of chaperones (such as Hsp70 or Grp78) in this manuscript but without discussing the potential of these surface forms in the development of therapeutic strategies. It is suggested to add the discussion of targeting surface form of chaperones by immunotherapy. Thanks for your remark. In the revised version, we discuss the potential therapeutic strategies aimed at HSP70 associated with the cell surface of CSCs (see blue text on page 19) and also at GRP78 localized to the cell surface of CSCs (see a separate subsection entitled ‘Targeting extracellular GRP78’ on pages 32-33, marked by blue).
Thanks for your consideration.
With best wishes,
On behalf of all the authors,
Alexander Kabakov
Round 2
Reviewer 1 Report
The authors have properly answered all concerns. I feel that the article has merits for its publication.
Reviewer 3 Report
I have checked revised version.
I have no comments.
My previous comments have been corrected.